# Application of human liver organoids as a patient-derived primary model for HBV infection and related hepatocellular carcinoma

Elisa De Crignis[1], Tanvir Hossain[1†], Shahla Romal[1†], Fabrizia Carofiglio[1], Panagiotis Moulos[2], Mir Mubashir Khalid[1], Shringar Rao[1], Ameneh Bazrafshan[1], Monique MA Verstegen[3], Farzin Pourfarzad[4], Christina Koutsothanassis[5‡], Helmuth Gehart[6], Tsung Wai Kan[1], Robert-Jan Palstra[1], Charles Boucher[7], Jan NM IJzermans[3], Meritxell Huch[8], Sylvia F Boj[4], Robert Vries[4], Hans Clevers[6], Luc JW van der Laan[3], Pantelis Hatzis[2], Tokameh Mahmoudi[1,9,10*]

[1]Department of Biochemistry, Erasmus University Medical Center, Rotterdam, Netherlands; [2]Biomedical Sciences Research Center 'Alexander Fleming', Vari, Greece; [3]Department of Surgery, Erasmus University Medical Center, Rotterdam, Netherlands; [4]Foundation Hubrecht Organoid Technology (HUB), Utrecht, Netherlands; [5]HybridStat Predictive Analytics, Kifisia, Greece; [6]Hubrecht Institute-KNAW, University Medical Centre Utrecht, Utrecht, Netherlands; [7]Department of Viroscience, Erasmus Medical Centre, Rotterdam, Netherlands; [8]Max Plank Institute of Molecular Cell Biology and Genetics, Dresden, Germany; [9]Department of Urology, Erasmus University Medical Center, Rotterdam, Netherlands; [10]Department of Pathology, Erasmus University Medical Center, Rotterdam, Netherlands

*For correspondence:
t.mahmoudi@erasmusmc.nl

†These authors contributed equally to this work

Present address: ‡Present address: Oxford Biodynamics, Oxford Business Park North, OX4 2JZ Oxford,, Oxford Biodynamics, Oxford Business, Oxford, United Kingdom

Competing interests: The authors declare that no competing interests exist.

**Abstract** The molecular events that drive hepatitis B virus (HBV)-mediated transformation and tumorigenesis have remained largely unclear, due to the absence of a relevant primary model system. Here we propose the use of human liver organoids as a platform for modeling HBV infection and related tumorigenesis. We first describe a primary ex vivo HBV-infection model derived from healthy donor liver organoids after challenge with recombinant virus or HBV-infected patient serum. HBV-infected organoids produced covalently closed circular DNA (cccDNA) and HBV early antigen (HBeAg), expressed intracellular HBV RNA and proteins, and produced infectious HBV. This ex vivo HBV-infected primary differentiated hepatocyte organoid platform was amenable to drug screening for both anti-HBV activity and drug-induced toxicity. We also studied HBV replication in transgenically modified organoids; liver organoids exogenously overexpressing the HBV receptor sodium taurocholate co-transporting polypeptide (NTCP) after lentiviral transduction were not more susceptible to HBV, suggesting the necessity for additional host factors for efficient infection. We also generated transgenic organoids harboring integrated HBV, representing a long-term culture system also suitable for viral production and the study of HBV transcription. Finally, we generated HBV-infected patient-derived liver organoids from non-tumor cirrhotic tissue of explants from liver transplant patients. Interestingly, transcriptomic analysis of patient-derived liver organoids indicated the presence of an aberrant early cancer gene signature, which clustered with the hepatocellular carcinoma (HCC) cohort on The Cancer Genome Atlas Liver Hepatocellular Carcinoma dataset and away from healthy liver tissue, and may provide invaluable novel biomarkers for the development of HCC and surveillance in HBV-infected patients.

## Introduction

Persistent hepatitis B virus (HBV) infection is the leading cause of chronic liver cirrhosis and hepato-cellular carcinoma (HCC) world wide (*MacLachlan and Cowie, 2015*; *Di Bisceglie, 2009*; *An et al., 2018*). A combination of viral and host factors determines whether an individual infected with HBV will be able to clear the infection or will become a chronic carrier. Characterized by its high host species and organ specificity, HBV infection and replication is thought to orchestrate an inter-play between the immune system and viral-specific factors that eventually lead to the onset of HCC. Insights into the molecular mechanisms underlying HBV-induced HCC have largely been provided by epidemiological studies (*El-Serag, 2012*; *Fattovich et al., 2008*; *Jiang et al., 2012*; *Sagnelli et al., 2020*), genome-wide analysis of viral and host characteristics (*Ally et al., 2017*; *Fujimoto et al., 2012*; *Huang et al., 2012*; *Ji et al., 2014*; *Sartorius et al., 2019*; *Shibata and Aburatani, 2014*; *Ally et al., 2017*), as well as by studies performed in in vitro settings using hepatoma cell lines (*Thomas and Liang, 2016*; *Zhang et al., 2014*).

However, the limited availability of relevant animals or in vitro model systems to study HBV infec-tion constitutes a major deficiency attributed to the strict viral host and cell-type tropism. Chimpan-zees remain the only animal model that supports the full HBV replication cycle, while available hepatoma cell line models are unsuitable for delineating the molecular steps leading to tumorigene-sis as they differ substantially from primary cells in their already tumor-derived gene expression pro-files (*Protzer, 2017*). These systems have inherent limitations resulting in poor predictive value for clinical outcomes (*Allweiss and Dandri, 2016*). Primary hepatocytes present the gold standard model system for HBV research in vitro, with recent studies reporting on significant increases in the half-life of these cultures (*Xiang et al., 2019*). However, primary cells are difficult to obtain and, since they cannot be expanded, are not usually available in quantities sufficient to perform large-scale analyses (*Hu et al., 2019*). Induced pluripotent stem cell (iPSC)-derived hepatocytes, which are susceptible to HBV infection and support replication, are also a useful model for studying host-virus determinants of replication (*Kaneko et al., 2016*; *Nie et al., 2018*; *Sakurai et al., 2017*; *Xia et al., 2017*). However, iPSC-derived hepatocyte models cannot be patient derived, confining studies to only ex vivo infection systems and limiting the possibility of patient-specific personalized treatment approaches (*Torresi et al., 2019*; *Nantasanti et al., 2016*). As a consequence of deficiencies in avail-able model systems and, despite its fascinating biology, many questions regarding the life cycle of HBV and its mechanisms of persistence, including HBV-induced molecular events underlying tumori-genesis, remain largely unexplored in primary settings, and key viral and host players involved remain unknown.

The dependence of hepatocytes on spatial and matrix-derived signals had until recently pre-vented their long-term in vitro culturing. Organoid culture technology involves the generation of cell-derived genetically stable in vitro 3D organ models of human origin. We have previously estab-lished a primary liver culture system based on isolation and expansion of primary cells that allows for the long-term expansion of liver cells as organoids (*Huch et al., 2013*; *Huch et al., 2015*). In this cul-ture system, isolated adult hepatic cells are expanded through multiple passages in an optimized expansion medium (EM) without induction of genomic alterations (*Huch et al., 2015*). When switched to a differentiation medium (DM) where proliferation signals are removed and ductal (pro-genitor) fate is inhibited, liver organoid cultures differentiate into functional hepatocytes in vitro as exemplified by their polygonal cell shape (*Figure 1A*) and hepatocyte functions, including *albumin* production and cytochrome *CYP3A4* expression and activity (*Huch et al., 2015*). Here we use the human liver organoid platform to model and study HBV infection and replication, as well as related tumorigenesis in patient-derived organoids generated from HBV-infected donors. This expandable model yields patient-specific organoids in quantities amenable to molecular and functional charac-terization and allows us to generate a living biobank of HBV-infected patient-derived cells amenable to downstream genomic, transcriptomic, and proteomic analysis as well as screening for HBV-directed therapeutics. We first described the ex vivo HBV infection of healthy donor (hD)-derived liver organoids, as a model to investigate viral infection and replication in hepatocytes. We used the HBV-infected organoid model as a platform for drug screening that can measure both drug-induced anti-HBV transcription and replication activity as well as drug-induced toxicity. We also

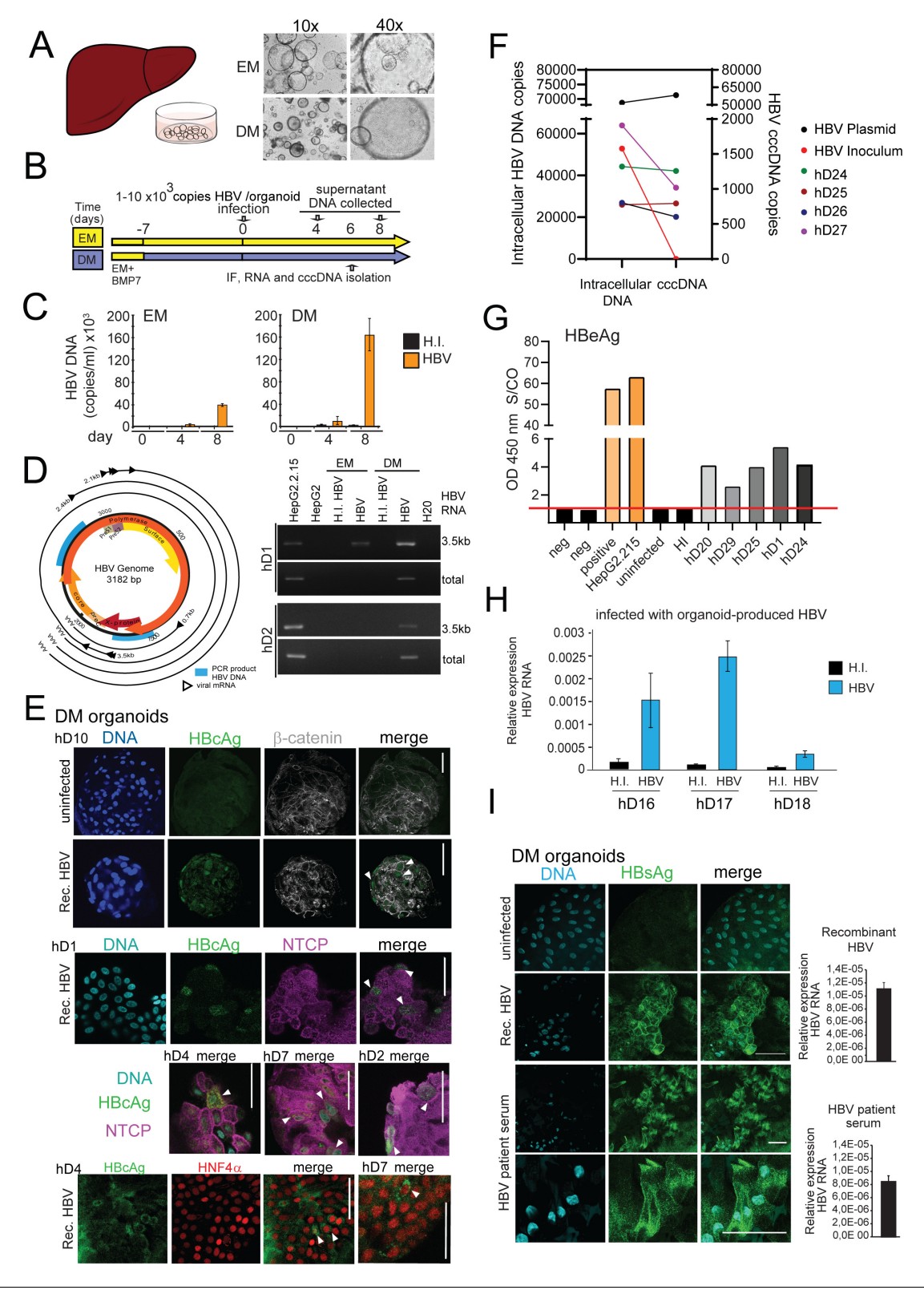

**Figure 1.** Modeling HBV infection in vitro using human liver organoids. (**A**) Representative images of liver organoids in expansion medium (EM) and differentiation medium (DM). (**B**) Experimental design of infection experiments. Arrows indicate the time points for hepatitis B virus (HBV) detection. (**C**) Levels of HBV DNA in supernatants of infected organoid cultures were quantified at indicated times by quantitative polymerase chain reaction (qPCR) and compared to the cultures challenged with heat-inactivated (HI) virus. (**D**) Schematic of the HBV genome showing open reading frames (ORFs)

*Figure 1 continued on next page*

*Figure 1 continued*

(arrows), HBV RNA transcripts (black circular lines) and the localization of PCR products (blue boxes). The agarose gel demonstrates expression of 3.5kb RNA transcript and total HBV RNA by nested PCR performed on complementary DNA (cDNA) obtained from two in vitro infected healthy donor (hD) organoid lines. (E) Immunofluorescent staining showing the expression of HBV core antigen (HBcAg) (green) together with sodium taurocholate co-transporting polypeptide (NTCP) (magenta), β-catenin (gray), or Hepatocyte Nuclear Factor 4 Alpha (HNF4α) (red) performed in different hD organoids 6 days after HBV infection in the differentiation medium. (F) Quantification of total intracellular HBV DNA (left Y-axis) and covalently closed circular DNA (cccDNA) (right Y-axis) purified from four hD organoid lines 6 days post infection. Quantification of total HBV DNA and cccDNA is also shown from HepG2.2.15-produced virus (inoculum, negative control for cccDNA) and from double-stranded HBV plasmid (as positive control for cccDNA) as indicated. (G) Quantification of HBV early antigen (HBeAg) was performed by enzyme-linked immunosorbent assay (ELISA) from the supernatant of infected organoids. Challenge with HI virus and uninfected organoids was used as a control for HBeAg present in the inoculum. Positive and neg bars correspond to positive and negative controls provided by the kit manufacturer. (H) Expression of intracellular HBV RNA relative to beta-2-microglobulin in three hD organoid lines infected with organoid-produced HBV (concentrated from pooled supernatants of organoid cultures infected with HepG2.2.15-produced HBV). (I) Immunofluorescent staining showing the expression of HBV surface antigen (HBsAg) (green) in DM organoids infected with recombinant HBV and patient serum. Scale bars represent 50 μm. Bar graphs show total HBV RNA levels in the culture at the time of staining. The online version of this article includes the following source data and figure supplement(s) for figure 1:

**Source data 1.** Source data for *Figure 1F*.

**Source data 2.** Source data for *Figure 1G*.

**Figure supplement 1.** Expression of HBV core Ag in infected human liver organoids.

**Figure supplement 2.** Release of HBV DNA declines over time in hD organoid lines infected in vitro.

**Figure supplement 2—source data 1.** Source data for *Figure 1—figure supplement 2C*.

**Figure supplement 2—source data 2.** Source data for *Figure 1—figure supplement 2D*.

**Figure supplement 3.** HBV-specific DNA, RNA, and proteins are detected in hD-derived organoids upon infection with recombinant HBV virus as well as with HBV-infected patient's serum.

demonstrated that transgenic modification of liver organoids provides an in vitro mechanistic platform to study the molecular determinants of HBV infection and replication. Finally, we performed transcriptomic analysis of HBV-infected patient-derived organoids and described the discovery of an early cancer gene signature, a potentially invaluable prognostic biomarker for HCC.

## Results

### Human liver organoids allow modeling of HBV infection in vitro

We first used the previously characterized liver organoid platform (*Huch et al., 2015*) on materials generated from healthy donors to set up a novel ex vivo HBV-infection system to study HBV replication. Liver organoids from healthy donors were grown in either EM or DM (*Figure 1A*) for 7 days prior to infection with recombinant HBV generated from HepG2.2.15, an HepG2 cell line subclone stably expressing HBV (*Figure 1B*). As control for the inoculum, we also infected organoids with heat-inactivated (HI) HBV. HBV infection and replication were validated by quantifying the levels of HBV DNA in the supernatant (*Figure 1C*), detection of intracellular HBV RNA (*Figure 1D*), visualizing HBV-specific proteins by immunofluorescence microscopy (*Figure 1E*), and quantifying intracellular covalently closed circular DNA (cccDNA) from infected organoids (*Figure 1F*). HBV DNA was detected in organoid culture supernatants from 4 days post infection, but not from the HI virus-infected cells, pointing to successful HBV replication (*Figure 1C*). Differentiated organoids maintained in DM were more efficiently infected and produced higher viral titers than organoids maintained in EM (*Figure 1C–D*). The RNA intermediates necessary for protein production and viral replication (3.5 kb RNA transcript and total HBV RNA) were present in infected DM organoids and detected by nested polymerase chain reaction (PCR) analysis, but not in the HI virus-infected cells (*Figure 1D*). As a further measure of active HBV replication, HBV early antigen (HBeAg) was also measured in supernatants of infected organoids and quantified (*Figure 1G*). Immunostaining, using antibodies recognizing HBV core antigen (HBcAg), showed specific nuclear and cytoplasmic staining in multiple infected healthy donor liver organoid lines, confirming the presence of foci of HBV replication in HBV-infected cells predominantly in infected DM organoids (*Figure 1E* and *Figure 1—figure supplement 1A–B*). Furthermore, infection of DM organoids resulted in the production of cccDNA, a definitive marker of HBV replication, as detected by a quantitative polymerase chain reaction (qPCR)-based cccDNA detection method of intracellular HBV DNA after digestion with a

nuclease to specifically remove non-cccDNA (*Figure 1F* and *Figure 1—figure supplement 1C*). Inoculum that lacks cccDNA was used as a negative control for the cccDNA-specific qPCR and HBV plasmid DNA was used as a positive control (*Figure 1F* and *Figure 1—figure supplement 1C*). HBV replication, infection, and spread appeared to be persistent until 8 days after infection when viral production dropped significantly, likely because of the limited half-life of differentiated organoids in culture (*Figure 1—figure supplement 2A*). Periodic culturing of the organoids in EM in order to stimulate the recovery and proliferation of the organoids modestly extended the half-life of the infected cultures, where viral production was maintained for approximately 1 month post infection (*Figure 1—figure supplement 2B*). Donor-specific differences in efficiency of HBV infection were observed consistent with variable HBV permissiveness of primary human hepatocytes (*Shlomai et al., 2014*), while HBV infection was observed with similar efficiency in different passages of the same donor line (*Figure 1—figure supplement 2C–D*). To determine whether the organoids are capable of producing infectious HBV, supernatants containing virus produced by organoids were collected, concentrated, and used for subsequent spinoculation of hD organoids. As shown in *Figure 1H*, infection of hD organoids with organoid-produced HBV resulted in expression of intracellular HBV RNA, indicating that organoids produce infectious viral particles. Growth of viral isolates from patient material has been limited by the lack of an adequate primary model system. However, differentiated organoids were able to support infection and replication when challenged with HBV-infected patient sera, as shown by production of viral DNA, expression of viral transcripts, and positive immunostaining for HBV surface antigen (HBsAg) (*Figure 1I* and *Figure 1—figure supplement 3*). Differentiated liver organoids therefore provide a useful ex vivo HBV-infection platform in which the role of specific host and viral factors can be investigated.

## Ex vivo HBV-infected liver organoids are a viable platform for anti-viral drug screening and drug-induced toxicity

We next examined whether the ex vivo infected liver organoid platform would be amenable to anti-HBV drug screening to monitor antiviral activity and drug-induced toxicity of two different drugs, tenofovir and fialuridine, according to the schematic outlined in *Figure 2A*. Tenofovir is a nucleoside reverse transcriptase inhibitor that inhibits the reverse transcription of HBV pre-genomic RNA to DNA. Fialuridine, also a nucleoside analog that inhibits reverse transcription, was shown to cause severe hepatotoxicity in patients (*McKenzie, 1995*). In the organoids, HBV viral DNA production in the culture supernatant was inhibited by both tenofovir and fialuridine in three independent hD-derived organoids, whereas, as expected, RNA levels remained the same (*Figure 2B*). Therefore, the organoid ex vivo infection platform not only allows measurement of drug-induced antiviral activity, but also offers insight into the mechanism of drug action by allowing delineation of distinct steps of the HBV life cycle targeted and inhibited. As expected, treatment of HepG2.2.15 cells with tenofovir and fialuridine resulted in similar decreases in released HBV DNA, but no change in intracellular HBV RNA levels (*Figure 2C*), reaffirming the mechanism of action of these drugs in a cell-line model of HBV replication. Due to the well-established detrimental effects of fialuridine on the viability of primary human hepatocytes, we sought to evaluate fialuridine-induced toxicity on primary human liver organoids as well as in HepG2 cells. We measured the viability of organoids and HepG2 cells using the alamarBlue viability assay and by monitoring their phenotype upon fialuridine and tenofovir treatment using microscopy. HepG2 cells demonstrated no change in cell viability upon treatment with tenofovir and increasing concentrations of fialuridine as compared to the mock-treated cells (*Figure 2D*). The phenotype of HepG2 cells was also comparable across all treatments as observed by microscopy (*Figure 2E*). Strikingly, the liver organoids treated with fialuridine at as low a concentration as 1 µM demonstrated a significant reduction in viability as measured by alamarBlue assay when compared to mock-treated cells (*Figure 2F* and *Figure 2—figure supplement 1A*). The organoids treated with higher fialuridine concentrations (5–20 µM) as well as with 20 µM tenofovir also demonstrated impaired viability (*Figure 2F* and *Figure 2—figure supplement 1A*). The decreased cell viability was also apparent in the phenotype of the 1–20 µM fialuridine-treated and 20-µM tenofovir-treated organoids compared to vehicle controls as observed by microscopy (*Figure 2G* and *Figure 2—figure supplement 1B*). This highlights that fialuridine-induced toxicity is evident and quantifiable in the primary human liver organoid model but not in the HepG2.2.15 model of HBV replication. Thus, we demonstrate that ex vivo-infected differentiated liver organoids support the full replication cycle of HBV and, following further characterization, may serve as an ideal novel primary

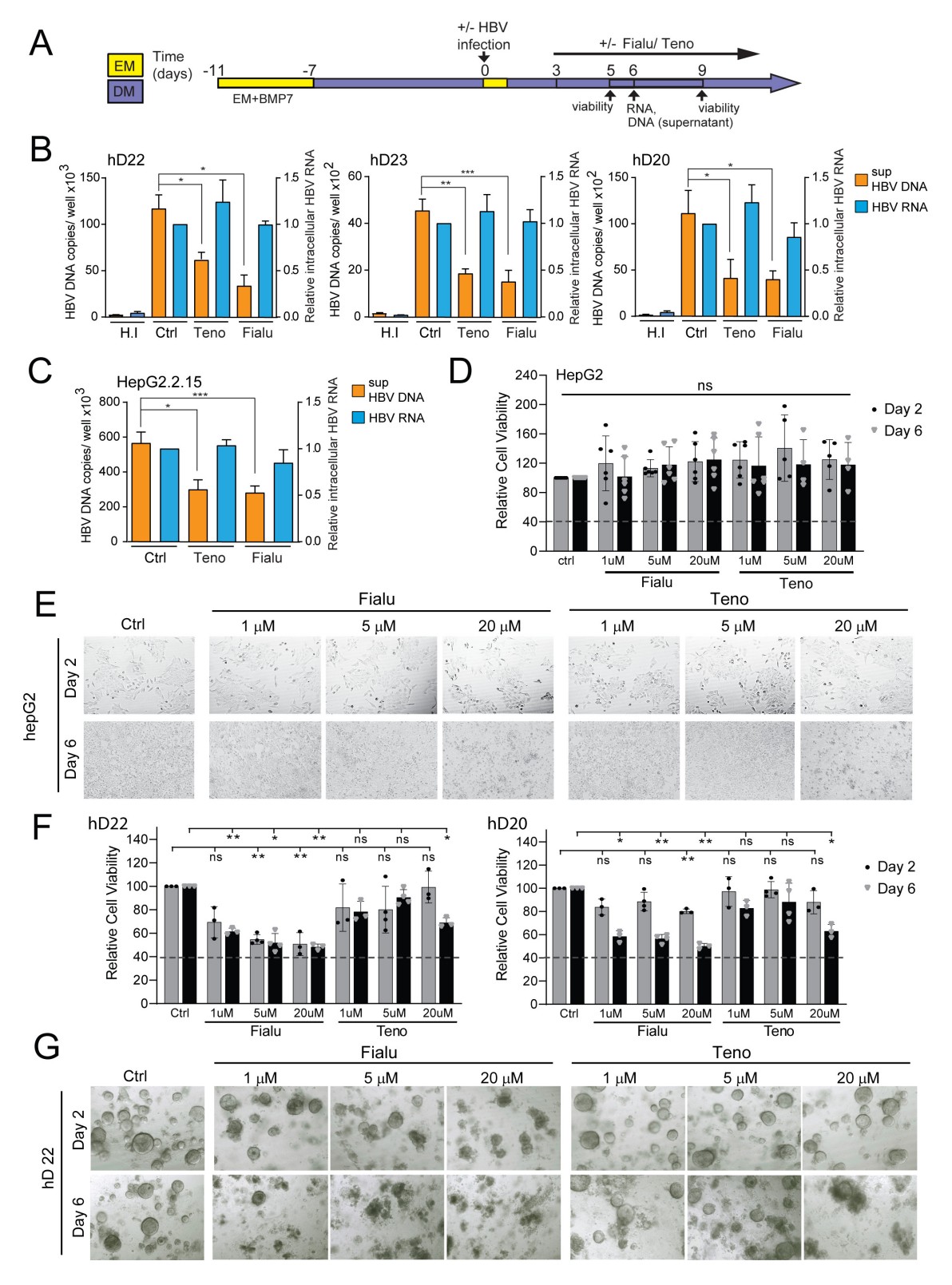

**Figure 2.** HBV-infected liver organoids as a model for HBV antiviral drug screening and toxicity. (**A**) Experimental design of drug treatment of hepatitis B virus (HBV)-infected liver organoids followed by assessment of antiviral activity and toxicity. Arrows indicate time points for HBV detection or assessment of viability. Levels of HBV DNA (orange) in the supernatant and intracellular HBV RNA (blue) (normalized to beta-2-microglobulin) were quantified by quantitative polymerase chain reaction (qPCR) and reverse transcription PCR (RT-PCR), respectively, for three independent healthy donors

*Figure 2 continued on next page*

*Figure 2 continued*

(B) and HepG2.2.15 cells (C) upon treatment with control vehicle, fialuridine (10 µM), or tenofovir (10 µM) as indicated. Data are shown as mean ± SD of at least three replicate treatments (paired two-tailed t-test); *p<0.05; **p<0.01; ***p<0.001. (D) Relative viability of HepG2 cells was measured using the alamarBlue cell viability assay after treatment with vehicle control, fialuridine, or tenofovir for 2 or 6 days as indicated, normalized to vehicle control, and plotted as the average of percent viability ± SD of at least four replicate treatments (paired two-tailed t-test) (ns = not significant). (E) Representative bright-field images taken of HepG2 cells treated with antiviral drugs for 2 or 6 days as indicated. (F) Bar diagrams representing relative cellular viability of healthy donor (hD) liver organoids after 2 or 6 days of treatment with fialuridine or tenofovir at the different concentrations indicated using the alamarBlue cell viability assay. All values are normalized to the vehicle-treated control and plotted as the average of percent viability ± SD of at least three replicate treatments (paired two-tailed t-test); *p<0.05; **p<0.01. The dotted line represents the lower limit of quantification based on values obtained from wells free of organoids containing the basement membrane matrix (BME) only. (G) Representative bright-field images taken of liver organoids treated for 2 or 6 days with the vehicle control or increasing concentrations of the antiviral drugs tenofovir or fialuridine as indicated.

The online version of this article includes the following source data and figure supplement(s) for figure 2:

**Source data 1.** Source data for *Figure 2B*.
**Source data 2.** Source data for *Figure 2C*.
**Source data 3.** Source data for *Figure 2D*.
**Source data 4.** Source data for *Figure 2F*.
**Figure supplement 1.** Human liver organoids as a model for anti-HBV drug-induced toxicity screening.

platform for drug screening as well as elucidation of the molecular events underlying HBV replication. Moreover, human liver organoids serve as an ideal platform for monitoring drug-induced toxicity in pre-clinical studies.

## HBV replication can be investigated in transgenically modified liver organoids

We observed higher HBV infection efficiency in DM organoids as compared to EM organoids (*Figure 1C*). This correlated with higher levels of sodium taurocholate co-transporting polypeptide (NTCP) expression, a cellular receptor expressed on the surface of hepatocytes implicated in HBV entry (*Yan et al., 2012*), in differentiated organoids as compared to organoids in EM (*Figure 3A–B*). HBV infection of liver organoids was dependent on NTCP as (pre)treatment with the competitive entry inhibitor myrcludex-B decreased infection of organoids as shown by quantitation of HBV intracellular RNA (*Figure 3C*) as well as HBV DNA (*Figure 3D*) and HBeAg (*Figure 3E*) in the supernatant of infected organoids. Exogenous expression of NTCP in hepatoma cell lines was shown to confer susceptibility to infection (*Yan et al., 2012*) in line with our observations of increased HBV infection in differentiated organoids, likely because of the higher level of NTCP expression. Since differentiated organoid cultures have a limited half-life, we sought to generate transgenically modified hD organoids exogenously expressing NTCP under expansion conditions (*Figure 3F–H* and *Figure 3—figure supplement 1A*) in order to improve infection efficiency and facilitate downstream analyses and investigation of the molecular events involved in HBV replication. We used a lentiviral construct harboring the coding sequence of Flag-tagged NTCP ubiquitously expressed under a Cytomegalovirus (CMV) promoter, followed by a blasticidin selection marker (*Figure 3F*). Immunofluorescence experiments performed on NTCP-liver organoids in the expansion phase confirmed high levels of NTCP protein expression correctly localized to the cellular membrane (*Figure 3H* and *Figure 3—figure supplement 1B*). Cholesterol target genes were induced in response to statin treatment in NTCP transgenic organoids, confirming the functionality of exogenously expressed NTCP (*Figure 3—figure supplement 1B–C*). We then evaluated viral production following HBV infection in transgenically modified NTCP organoid lines as compared to parental lines (*Figure 3I–J*). Interestingly, comparable levels of HBV DNA and HBsAg were observed in the supernatants of both parental and NTCP-expressing organoid lines (*Figure 3I–J*), suggesting that expression of NTCP alone is not sufficient to improve HBV infection rate in liver organoids grown in EM (*Figure 3I–J* and *Figure 3—figure supplement 1D*).

To further highlight the ability of liver organoids to be transgenically modified, we produced a long-term, expandable, primary HBV-producing liver organoid model system that can be used to study HBV transcription events. We utilized a lentiviral construct to produce transgenic organoid lines containing an integrated copy of HBV (*Figure 3—figure supplement 2*). 'Lenti-HBV' (*Figure 3—figure supplement 2C–D*) allowed for the generation of replication-competent, expandable, long-

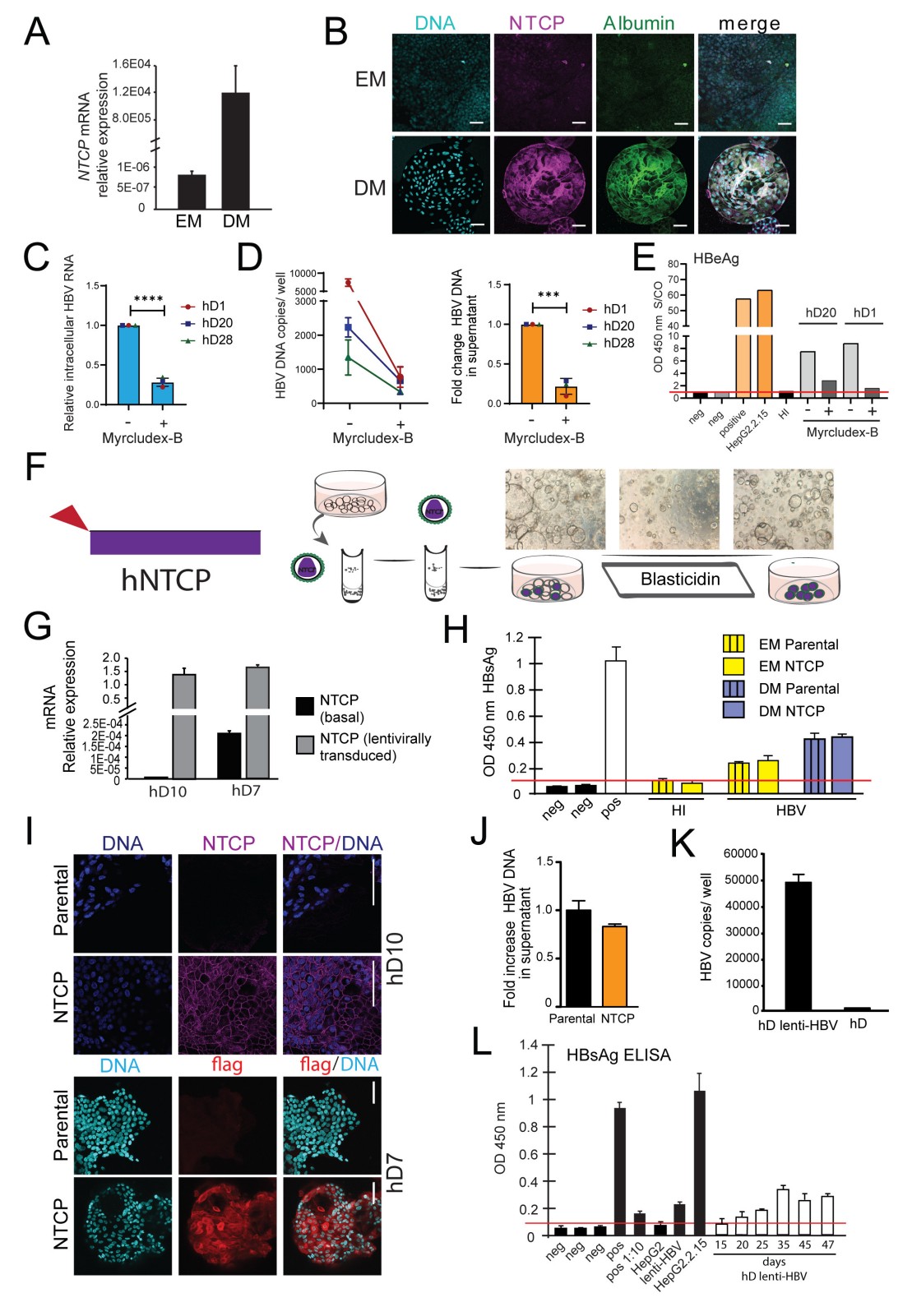

**Figure 3.** Lentivirally transduced transgenic liver organoids in the study of HBV. (**A**) mRNA expression levels of the hepatitis B virus (HBV) receptor sodium taurocholate co-transporting polypeptide (NTCP) in undifferentiated (EM) and differentiated (DM) organoids (n = 3); mRNA levels were calculated according to the 2ΔCt method using *GAPDH* as the reference gene. (**B**) Immunofluorescent staining showing the expression of NTCP (magenta) in EM and DM organoids. Nuclei were counterstained with Hoechst 33343 (cyan). Scale bars represent 50 μm. (**C–E**) Inhibition of HBV

*Figure 3 continued on next page*

*Figure 3 continued*

infection of liver organoids by (pre)treatment with myrcludex-B (10 µM). Control or myrcludex-B-treated differentiated liver organoids were infected with HBV, and intracellular HBV RNA (C), copy numbers of HBV DNA genome in the culture supernatant (D), and HBV early antigen (HBeAg) (E) produced in the culture supernatant were determined 5–6 days after inoculation. Error bars represent mean ± SD from three independent donors (unpaired two-tailed t-test); ***p<0.001; ****p<0.0001. (F) Schematic representation of the experimental procedure for the transduction experiments. Following infection with a lentiviral vector expressing Flag-NTCP, organoids were selected with blasticidin for 5 days in order to obtain lines expressing NTCP in the expansion phase. (G) Levels of expression of NTCP were evaluated by reverse transcription polymerase chain reaction (RT-PCR) in the untransduced (parental) and the transduced (NTCP) lines. Expression of NTCP was calculated according to the 2ΔCt method using the housekeeping gene *GAPDH* as the reference gene and confirmed by immunofluorescence staining targeting NTCP (magenta) or Flag (red) (I). (H) HBV surface antigen (HBsAg) released in the supernatant of parental and NTCP organoid lines grown in EM or DM 10 days after HBV infection was detected by enzyme-linked immunosorbent assay (ELISA). Challenge with heat-inactivated virus was used to control for HBsAg present in the inoculum. Pos and neg bars correspond to positive and negative controls provided by the kit manufacturer. Threshold for positivity (red line) was calculated as the average OD + 2SD of negative controls. (J) HBV DNA in the supernatant of NTCP-expressing organoid cultures was quantified 5 days after infection and compared to DNA detected in the supernatant of untransduced HBV-infected organoids (n = 3). Bars represent the fold increase in HBV DNA detected in the supernatant, and untransduced HBV-infected organoids were used as the reference. Relative amounts of (K) HBV DNA and (L) HBsAg produced by healthy donor (hD) lenti-HBV organoid lines.

The online version of this article includes the following source data and figure supplement(s) for figure 3:

**Source data 1.** Source data for *Figure 3C*.
**Source data 2.** Source data for *Figure 3D*.
**Source data 3.** Source data for *Figure 3E*.
**Figure supplement 1.** hD organoid lines expressing high levels of functional NTCP can be generated by lentiviral infection.
**Figure supplement 2.** Generation of a lentiviral construct used to establish human liver organoid lines expressing viral RNA and releasing virus in the supernatant.

term, hD transgenic lenti-HBV organoid lines, in which transcription from the HBV transgene results in the production of viral proteins and particles (*Figure 3K–L*, *Figure 3—figure supplement 2D*). This model system provides a primary platform to screen for inhibitors of HBV transcription and is a primary-cell alternative to the HepG2.2.15 cell lines that produce very low levels of cccDNA (*Sells et al., 1988*), for studies into HBV pathogenesis. Thus, the amenability of human liver organoids to transgenic modification enables investigation of HBV replication and in-depth characterization of the molecular events involved.

## Characterization of non-tumor HBV-infected patient-derived liver organoids

The ability to generate a patient-derived primary model is a key advantage of using the liver organoid platform. We applied the previously characterized method to generate liver organoids from healthy donors (*Huch et al., 2015*) to generate novel patient-derived organoids from HBV-infected individuals undergoing liver transplantation (*Figure 4A*). The explant used for generating patient-derived organoids was HBV-infected, chronically cirrhotic liver tissue obtained from 18 explanted livers from people infected with HBV (*Figure 4A and B*, *Table 1*). We generated and expanded organoid cultures from fresh and frozen explant tissue from all donors with similar efficiency (data not shown).

Non-tumor HBV-infected patient-derived (infected patient [iP]) organoids were expanded in culture (EM) and displayed proliferation rates (*Figure 4B* and data not shown) and expression of progenitor markers *LGR5, KRT7, HNF4α*, and *Sox9* (*Figure 4C*) comparable to those of hD organoids. When grown in DM, in which proliferation signals are removed and the progenitor fate is inhibited, liver organoid cultures from both healthy and HBV-infected sources acquired the hepatocyte fate and differentiated into functional hepatocytes, similar to previously obtained data for healthy liver organoids (*Huch et al., 2015*). In DM, both hD and iP liver organoid cultures showed increased expression of hepatocyte-specific genes encoding *albumin*, cytochrome *CYP3A4*, and NTCP, the cellular receptor implicated in HBV hepatocyte entry, concomitant with decreased expression of the stem cell-specific gene *LGR5* (*Figure 4D*). None of the iP organoids showed signs of HBV production at the RNA, DNA, or protein level (data not shown), suggesting that adult stem cells, from which the liver organoids are derived, do not contain actively replicating HBV. Interestingly, five out of six iP organoids contained portions of integrated HBV, corresponding to the HBx coding region (*Figure 4—figure supplement 1*). In accordance with our findings, in previous genome-wide analyses in

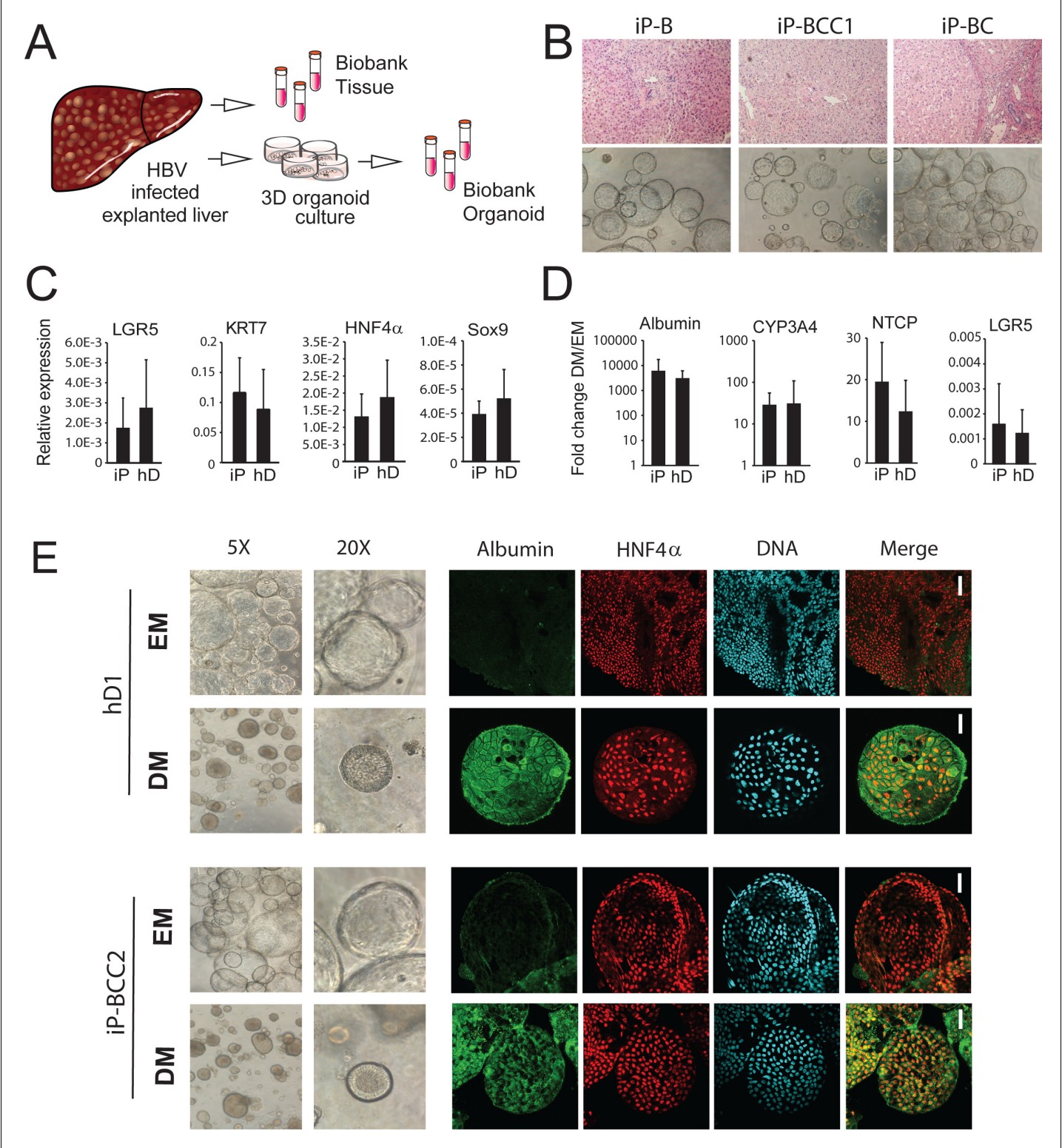

**Figure 4.** Characterization of organoid cultures from liver explants of HBV-infected patients. (**A**) Representative panel showing the procedure to generate organoid cultures or biobanks from liver tissue. (**B**) Hematoxylin-eosin-stained sections of explanted liver tissue and phase-contrast pictures showing the morphology of liver organoids derived from hepatitis B virus (HBV)-infected individuals. (**C**) Expression profile of the progenitor markers *LGR5, KRT7, HNF4α*, and *Sox9* in EM (undifferentiated) organoids derived from liver of healthy donors (hD) (n = 4) and HBV-infected individuals (iP) (n = 5). Levels of expression were calculated according to the 2ΔCT method using *GAPDH* as the reference gene. (**D**) Differentiation capacity of organoid cultures derived from liver of hDs (n = 4) and iPs (n = 5). Bars represent the fold difference in the expression of hepatocyte-specific genes

*Figure 4 continued on next page*

*Figure 4 continued*

encoding *albumin*, cytochrome *CYP3A4*, and sodium taurocholate co-transporting polypeptide (NTCP), and the progenitor-specific gene *LGR5* in DM (differentiated) cultures compared to EM organoids using the 2ΔΔCT method. (**E**) Immunofluorescent staining targeting albumin (green) and HNF4α (red) was performed in EM and DM organoids. Phase-contrast images, depicting the morphology of the cells, are shown as reference.

The online version of this article includes the following source data and figure supplement(s) for figure 4:

**Figure supplement 1.** Parts of HBV genome are found integrated in patient-derived liver organoids.
**Figure supplement 1—source data 1.** Source data for *Figure 4—figure supplement 1B*.
**Figure supplement 1—source data 2.** Source data for *Figure 4—figure supplement 1C*.
**Figure supplement 2.** iP-derived and hD-derived organoids have comparable differentiation potential.

tissues derived from donors with HCC, this region of the HBV genome was also found to be highly integrated in both HCC and non-tumor hepatocytes from infected patients (*Sung et al., 2012*; *Zhao et al., 2016*). Interestingly, in one donor, who was previously infected with HBV but cleared the infection (iDonor1), integration corresponding to a part of the HBV genome, including the HBx

**Table 1.** Characteristics of patients included in the study.

| Patient | Age | Sex | Diagnosis | Treatment (Y/N) | HBsAg (P/N) | HBcAg (P/N) | HBeAg (P/N) | Anti-HBsAg | Anti-HBcAg | Anti-HBeAg | DNA |
|---|---|---|---|---|---|---|---|---|---|---|---|
| iDonor 1 | 44 | F | DONOR, cleared HBV | N | N | NA | NA | P | P | NA | NA |
| iDonor 2 | 48 | F | DONOR, cleared HBV | N | N | NA | NA | P | P | NA | NA |
| iDonor 3 | 62 | M | DONOR, cleared HBV | N | N | NA | NA | N | P | NA | NA |
| iDonor 4 | 56 | M | DONOR, cleared HBV | N | N | NA | NA | P | P | NA | NA |
| iP-BC | 49 | M | Liver cirrhosis based on chronic HBV infection | Y | P | NA | NA | P | P | NA | <20 |
| iP-BCC1 | 65 | M | HBV liver cirrhosis with HCC | Y | P | NA | NA | N | P | NA | NA |
| iP-BCC2 | 54 | M | Chronic liver cirrhosis along with HCC | Y | P | NA | NA | N | P | NA | <20 |
| iP-BCC3 | 68 | M | Chronic HBV liver cirrhosis with HCC | Y | P | NA | NA | N | P | NA | <20 |
| iP-BCC4 | 52 | M | HBV liver cirrhosis with multiple HCC | Y | P | NA | N | N | P | P | <20 |
| iP-BCC5 | 51 | M | HBV liver cirrhosis with HCC | Y | P | NA | NA | N | P | NA | <20 |
| iP-BCC6 | 63 | M | HBV liver cirrhosis with HCC | Y | P | N | N | N | P | P | <20 |
| iP-BCC7 | 61 | M | HBV liver cirrhosis with HCC | Y | P | NA | NA | N | P | NA | 28 |
| iP-BCC8 | 53 | M | HBV liver cirrhosis with HCC | Y | P | NA | N | N | P | P | <20 |
| iP-BFA | 54 | F | Acute liver failure based on HBV with Crohn's disease | N | P | NA | N | N | P | P | <20 |
| iP-BFA2 | 62 | M | Acute liver failure based on HBV | N | P | NA | NA | P | P | NA | NA |
| iP-BDFA | 28 | M | Acute liver failure based on HBV and HDV co-infection | N | P | NA | N | NA | P | P | 1180 |
| iP-BDFA2 | 60 | F | Acute liver failure based on HBV and HDV co-infection | Y | P | NA | NA | N | P | NA | 1080 |
| iP-BDCC | 59 | M | HCC, HBV co-infected with HDV | Y | P | NA | NA | N | P | P | <20 |

| Patient | Age | Sex | Diagnosis | Treatment (Y/N) | HBsAg (P/N) | HBcAg (P/N) | HBeAg (P/N) | Anti-HBsAg | Anti-HBcAg | Anti-HBeAg | DNA | Nash (Y/N) |
|---|---|---|---|---|---|---|---|---|---|---|---|---|
| CN1 | 69 | F | Nash-HCC | N | N | N | N | N | N | N | NA | Y |
| CN2 | 71 | M | Nash-HCC | N | N | N | N | N | N | N | NA | Y |
| CN3 | 66 | F | Nash-HCC | N | N | N | N | N | N | N | NA | Y |
| CN4 | 60 | F | Nash-HCC | N | N | N | N | N | N | N | NA | Y |

region, was also observed (*Figure 4—figure supplement 1*). This finding indicates that despite clearance of active HBV replication, portions of the genome may remain integrated in the hepatocyte genome, for which the functional consequences remain to be elucidated. At the phenotypic level, while EM organoids grew larger in size and were translucent, differentiated iP organoids showed hepatocyte morphology and physiology comparable to hD organoids, with thickening of the outer cell layer, comparable to hD organoids (*Figure 4E* and *Figure 4—figure supplement 2A*) and production of albumin as detected by immunofluorescence staining (*Figure 4E* and *Figure 4—figure supplement 2B*). Thus, non-tumor HBV iP organoids display phenotypes comparable to hD-derived organoids, retain the capacity for differentiation, and are conducive to downstream genomic, transcriptomic, and proteomic analysis.

## HBV-infected patient-derived liver organoids display a distinct early gene expression signature

The early detection of HBV-related HCC is a challenge that remains critical to direct optimal clinical management of the disease. Despite widely practiced periodic surveillance of patients with cirrhosis, patients with HCC are mostly diagnosed in a late stage. The presence of diagnostic biomarkers for early events in liver cell tumorigenesis would therefore be invaluable for early detection. In order to identify potential early biomarker genes for HBV-induced HCC, we performed mRNA sequencing of the organoid lines derived from HBV-iP and compared their gene expression profiles to those of organoids derived from hDs. We performed hierarchical clustering of protein-coding differentially expressed genes obtained from the comparison between organoids from five hD organoid lines and those from five iP lines (*Figure 5A*, *Table 1*). The iP organoid lines were seeded from five HBV mono-infected patients with cirrhotic liver, four of whom presented with small tumors at the time of the explant (*Figure 5A*, *Table 1*). Interestingly, although the hD and iP organoids were phenotypically indistinguishable, the iP organoids clustered separately from the healthy donors (*Figure 5A*). This comparison resulted in identification of a gene signature characteristic of iP-derived organoids (henceforth 'iP-signature') (*Figure 5A* and *Supplementary file 1*). We then ranked the differentially expressed genes according to the relative distance of their expression in hD- vs iP-derived organoids and identified a group of 33 putative early biomarker genes (*Figure 5B*). Gene ontology (GO)-term and KEGG-pathway analysis revealed that the 'early signature' genes were enriched in metabolic pathway-associated genes (*Supplementary file 2*). Among these, *CCNA1* and *STMN2*, which we found to be upregulated in the non-tumor infected patient-derived organoids, were previously identified to be upregulated in HCC (*Allain et al., 2016*; *Chen et al., 2019*; *Gao et al., 2008*; *Paradis et al., 2003*). Conversely, *WNK2, RUSC2, CYP3A4,* and *RGN*, among the significantly downregulated genes in the non-tumor iP organoids, have been described as tumor suppressors downregulated in HCC (*Allain et al., 2016*; *Ashida et al., 2017*; *Tao et al., 2011*; *Yamaguchi, 2015*). We further checked if these targets were also upregulated in primary hepatocytes obtained from the tissues from which iP vs hD organoids had been seeded (*Figure 5C*) or in ex vivo HBV-infected organoids (*Figure 5—figure supplement 1*). Indeed, we observed upregulation of mRNA expression of the organoid-derived signature genes *ADAMTS1, CORO1A, PTHLH, STMN2,* and *STY5* in primary hepatocytes obtained from HBV-infected patients as compared to those obtained from healthy donors (*Figure 5C*). Similarly, signature genes were upregulated upon ex vivo HBV infection of hD organoids (*Figure 5—figure supplement 1*). Therefore, transcriptomic analyses of healthy vs patient-derived organoids resulted in the identification of an HBV infection early gene signature and possible biomarkers for HBV infection.

## Transcriptomic analysis of HBV-patient-derived organoids results in identification of cancer gene signature

We then applied the iP-signature derived from our analysis (*Figure 5A*) to all our liver organoid samples. *Table 1* describes all patients, groups, and clinical characteristics. Our subsequent analysis included six healthy donors (orange circles, hD), six HBV-infected samples (green triangles, iP), and four healthy donors that had previously been infected with HBV, but had subsequently cleared infection and displayed no aberrant phenotype (blue squares, infected donor [iDonor]) (*Figure 6A and B*). The application of the iP-signature to these samples groups hDs (gray bars) apart from iPs (black bars), as expected (*Figure 6A*). Interestingly, the organoids generated from the iDonors (light

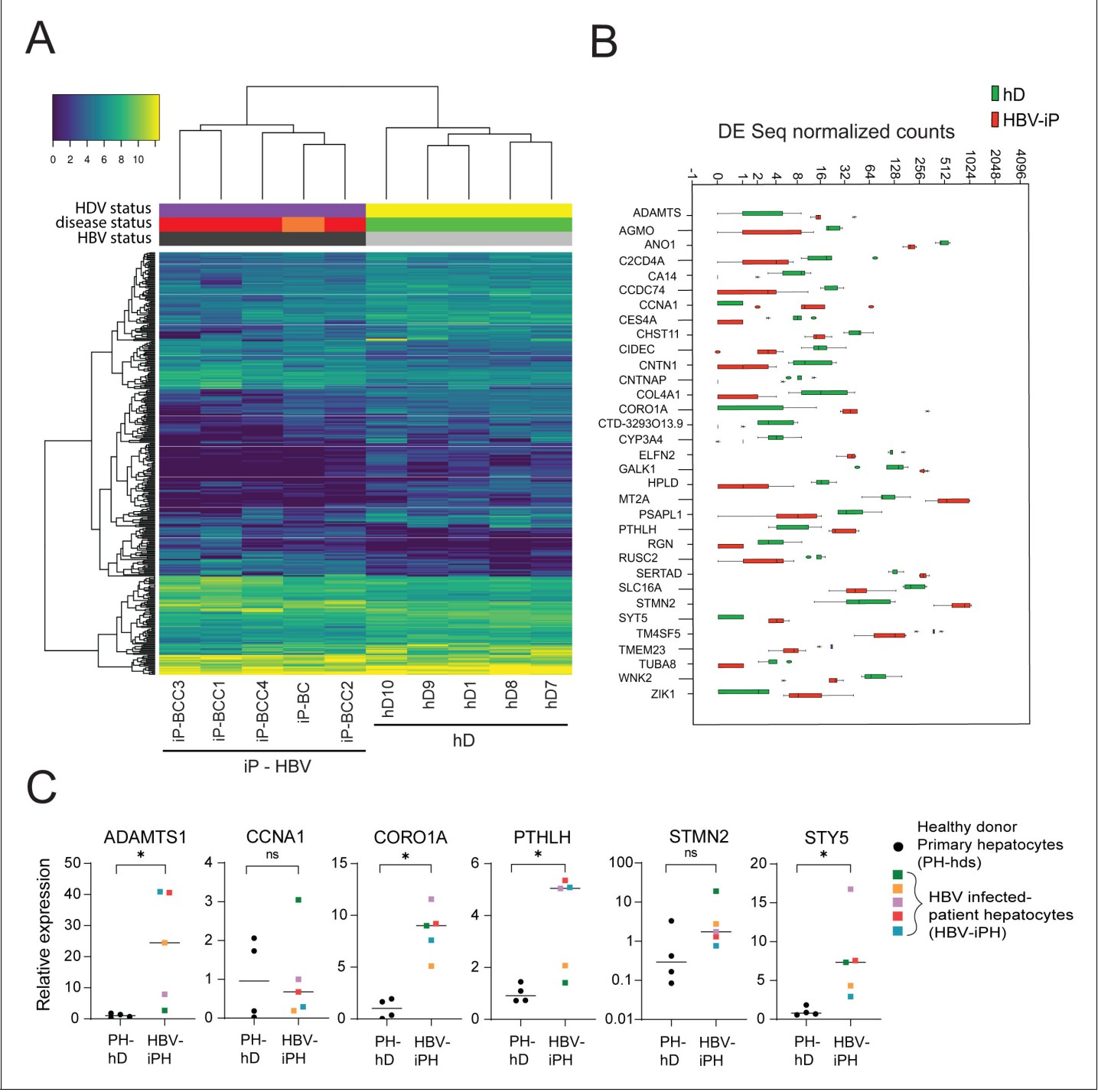

**Figure 5.** Differential gene expression in patient-derived liver organoids. (**A**) Hierarchical clustering heatmap of differentially expressed genes derived from the comparison between the group of five healthy donors (hDs) and five infected patients (iPs) presenting hepatitis B virus (HBV) infection and cirrhosis (all) and hepatocellular carcinoma (HCC) (four out of five). (**B**) Box plot of DESeq normalized counts of 33 putative biomarker genes obtained from hD organoids (depicted in green) or HBV-iP organoids (depicted in red). (**C**) Relative expression of mRNAs of signature genes *ADAMTS1, CCNA1, CORO1A, PTHLH, STMN2*, and *STY5* in hD primary hepatocytes (n = 4, black circles) and in hepatocytes from HBV-infected patients (n = 5, each donor with its own colored square) was quantified by reverse transcription polymerase chain reaction (RT-PCR) using *cyclophilin A* mRNA as an internal control. All expression values are normalized to the average mRNA expression of the hDs (unpaired two-tailed Mann-Whitney test); ns = not significant; *p<0.05.

The online version of this article includes the following source data and figure supplement(s) for figure 5:

**Source data 1.** Source data for *Figure 5C*.

*Figure 5 continued on next page*

*Figure 5 continued*

**Figure supplement 1.** Upregulation of signature genes in ex vivo HBV infected liver organoids.

**Figure supplement 1—source data 1.** Source data for *Figure 5—figure supplement 1A*.

orange bars), which had cleared HBV infection, clustered together and more closely to the larger group of iP organoids derived from patients with HBV-driven cirrhosis and HCC (maroon bars) (*Figure 6A*). This observation of iDonors' clustering closely with iP-HBV-HCC indicated the presence of the early HCC-like gene signature in these infected donor livers, despite clearance of HBV infection and absence of phenotypic and functional abnormalities at the time of donation, an observation that may be important to be considered for transplantation purposes and surveillance. This is also in accordance with our observation that some iDonors also contain integrated parts of the HBV genome corresponding to HBx (*Figure 4—figure supplement 1*). We then applied the iP-signature to organoids from 11 iPs (green triangles, iP), 6 healthy donors (orange circles, hD), 4 infected donors (blue squares, iDonors), and 4 donors diagnosed with non-alcoholic steatohepatitis (NASH) not caused by HBV infection (purple pentagons, CN) (*Figure 6C–D*). *Table 1* describes all patients, groups, and clinical characteristics. Included amongst the iP group were also organoids seeded from HBV-infected patients who at the time of collection had HCC (iP-BCC), acute liver failure (iP-BFA), HBV-infection-induced cirrhosis without HCC (iP-BC), as well as HBV-HDV coinfection either with HCC (iP-BDCC) or with acute liver failure (iP-BDFA). The iP-signature separates and groups together the non-infected samples (gray bars) and iP organoids (black bars) (*Figure 6C*). Importantly, although the CN samples were obtained from cirrhotic livers (green bars), they grouped apart from the HBV-infected iPs and hDs (*Figure 6C–D*), indicating that our gene signature is defined by HBV-infection status, rather than the cirrhotic status. In agreement, multidimensional scaling analysis of all iP and hD samples indicated that all hD organoids are clearly separated from the rest and tightly grouped together, while the iP organoids form separate groups corresponding to HDV (hepatitis D virus) coinfection, HCC, acute liver failure, or previous/cirrhotic HBV infection and cluster together, but away from CN samples (*Figure 6A–D*). We have, therefore, identified a specific gene signature/ group that discriminates between the hD and iP organoids among more complex classifications (*Figure 6A–D*). We applied this gene signature to all samples in The Cancer Genome Atlas Liver Hepatocellular Carcinoma (TCGA-LIHC) database, a depository of sequences from HCC patients. The application of this gene signature when mapped onto the relevant TCGA gene expression data distinctly separated the TCGA HCC samples from non-HCC samples (*Figure 6E*). This important observation indicates that transcriptomic analysis of our patient-derived liver organoid model identified a novel early liver cancer gene signature in non-tumor HBV-infected patient-derived liver organoids, despite the absence of phenotypic signs of aberrant growth. Thus, HBV-infected patient-derived liver organoids are a novel primary 3D cell culture model that resembles the diseased tissue of origin and can be used for genomic, transcriptomic, proteomic, and clinical applications to identify biomarkers for disease states during HBV infection.

## Discussion

Human liver organoids are a novel in vitro primary model system that supports HBV infection and replication, can be utilized as a platform to study HBV pathogenesis as well as for potential anti-HBV-drug screening, can be transgenically modified for downstream mechanistic and functional assays, and can be seeded from HBV-infected patient livers for personalized molecular and functional analyses (*Figure 7*). Healthy donor liver organoids were efficiently infected with both recombinant virus as well as HBV-infected patient serum, expressed HBsAg, HBeAg, and HBV core proteins, produced cccDNA and infectious HBV in the culture supernatant, indicating that liver organoids support the full HBV replication cycle (*Figure 1*). Human liver organoids, therefore, serve as an invaluable tool in the field of HBV research to investigate the molecular mechanisms underlying HBV replication in a primary cell system that is expandable and ideal for biobanking.

We demonstrated that the ex vivo infected organoid replication platform is amenable to screening for potential inhibitors of HBV replication and can provide insights into both HBV-directed antiviral activity and drug-induced toxicity. The ability to measure different readouts of HBV replication

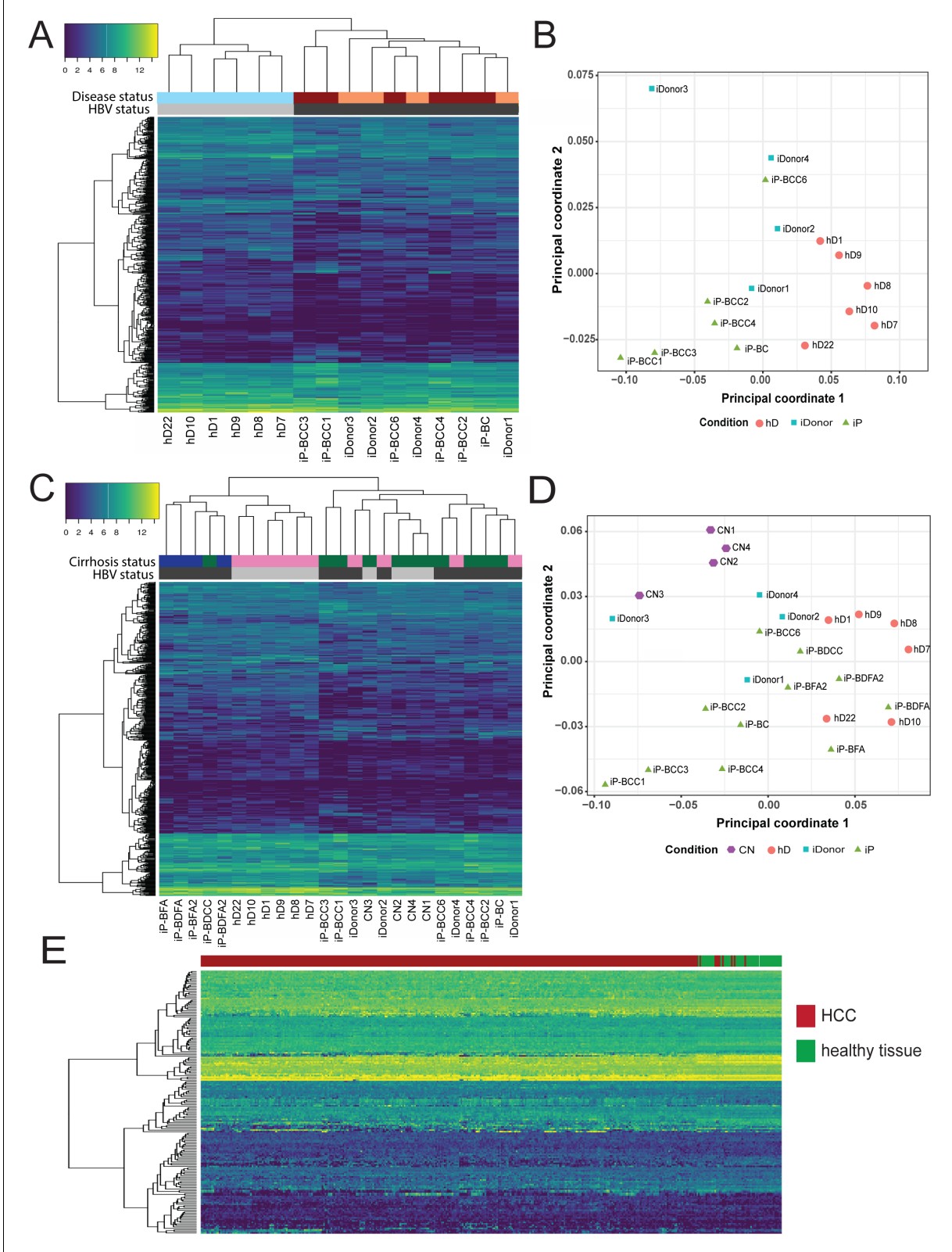

**Figure 6.** Characterization of early HCC gene signature. (**A**) Hierarchical clustering heatmap of six infected patient (iP), six healthy donor (hD), and four infected donor (iDonor) samples depicting the grouping and expression levels of protein-coding differentially expressed genes derived from the comparison between the group of five hDs and five iPs presenting hepatitis B virus (HBV) infection, cirrhosis, and hepatocellular carcinoma (HCC) (four out of five) (from *Figure 5A*). Colored bars on top of the heatmap indicate HBV status (black for HBV-positive patients, gray for healthy donors) and

*Figure 6 continued on next page*

*Figure 6 continued*

disease status (maroon for HBV-positive cirrhosis and HCC, orange for healthy donors that have cleared HBV infection, sky blue for healthy donors that have not been infected with HBV). (**B**) Multidimensional scaling plot of iP (green triangles), hD (orange circles), and iDonor samples (blue squares). (**C**) Hierarchical clustering heatmap of all 11 iP, 6 hD, 4 CN, and 4 iDonor samples depicting the grouping and expression levels of protein-coding differentially expressed genes derived from the comparison between the group of five hDs and five iPs presenting HBV infection and HCC (four out of five) (from *Figure 5A*). Colored bars on top of the heatmap indicate HBV status (black for HBV-infectected patients and donors, gray for non-HBV infected healthy donors and CN patients ) and cirrhosis status (green for cirrhotic livers, blue for acute HBV-infected non-cirrhotic livers, and pink for non-cirrhotic livers from both healthy and infected donors). (**D**) Multidimensional scaling plot of iP (green triangles), hD (orange circles), CN (purple pentagons), and iDonor samples (blue squares). (**E**) Hierarchical clustering heatmap of liver HCC gene expression data from 342 HCC tissue samples and 47 samples from matched nearby tissue from The Cancer Genome Atlas (TCGA) using the HBV-iP vs hD gene signature (from *Figure 5A*).

at the level of HBV mRNA gene expression, or viral production by quantitation of HBV DNA or S

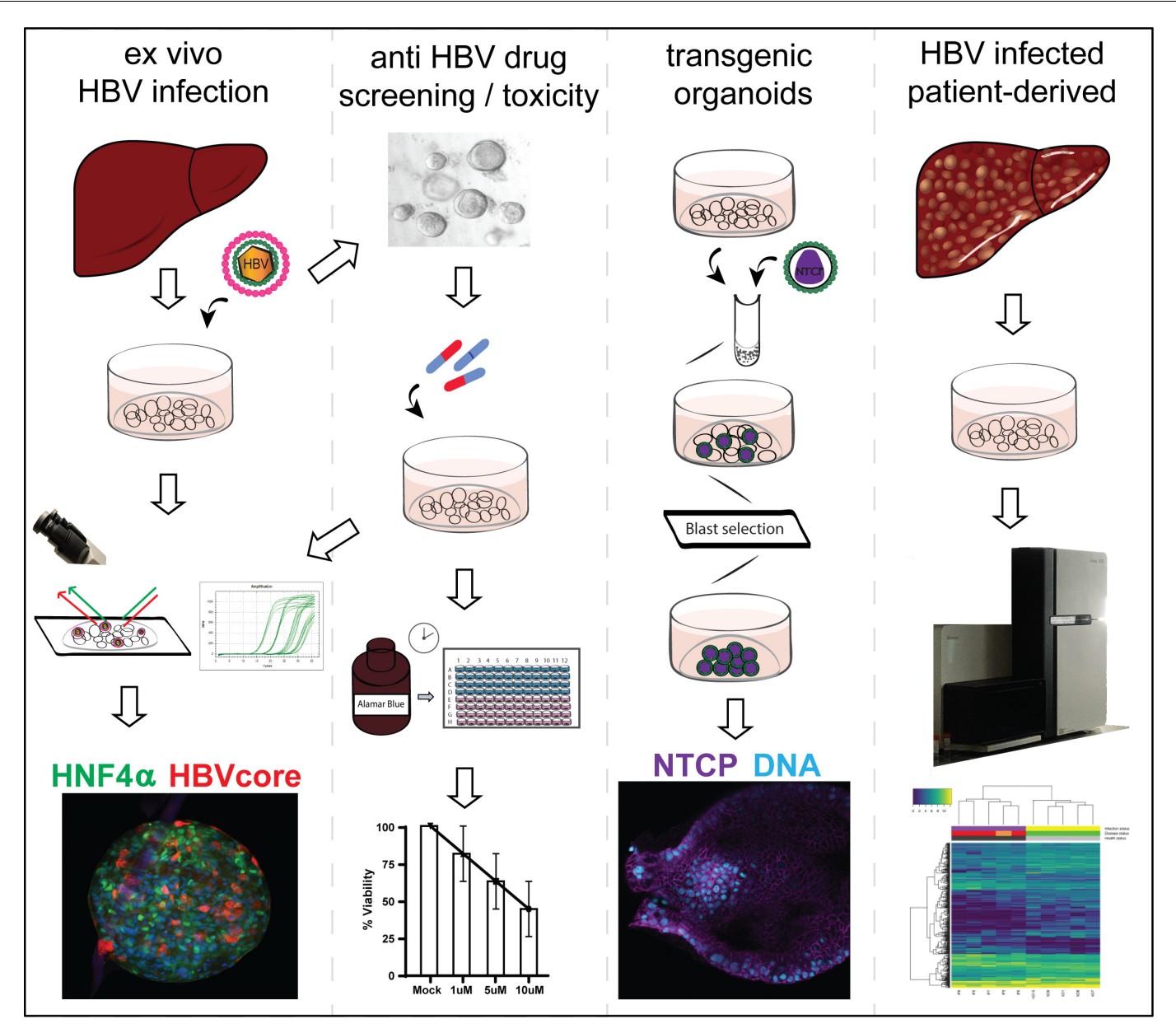

**Figure 7.** Schematic of applications of human liver organoids in HBV studies.

antigen, present in the culture supernatant of HBV-infected liver organoids, can provide insights into the distinct steps in the HBV life cycle that may be targeted upon drug treatment. This is highlighted by our studies using the HBV-directed drugs tenofovir and fialuridine (*Figure 2A*), which, as expected, did not affect the levels of intracellular HBV RNA, but blocked HBV replication by inhibiting the reverse transcription step that converts HBV pre-genomic RNA to DNA, as was measurable by decreased HBV DNA levels. A critical aspect of potentially therapeutic novel drugs, which leads to failure within the drug development pipeline, is toxicity. While fialuridine treatment of HBV-infected human liver organoids efficiently inhibited HBV replication, it also caused significant toxicity in the liver organoids. The potential of this pre-clinical human liver organoid model for measuring drug-induced toxicity is highlighted by the results of a phase II clinical trial conducted in 1993 to evaluate fialuridine as a novel HBV-anti-viral drug (Eli Lilly Trial H3X-MC-PPPC, NIH protocol #93-DK-0031). Although the preliminary results of the trial were promising, with reduced levels of HBV DNA in patients, the study was terminated on an emergency basis by week 13 because of serious hepatotoxicity in seven patients, of which five died and two required emergency liver transplantation (*McKenzie, 1995*). This compound successfully passed pre-clinical toxicity studies, but the expression of a nucleoside transporter in human mitochondria may be responsible for the human-specific mitochondrial toxicity caused by fialuridine (*Lee et al., 2006*). In the primary human liver organoid system, the toxicity of fialuridine is significant and quantifiable (*Figure 2E–F*) but was not observed in the human HepG2 hepatoma cell line, indicating that human liver organoids can more accurately predict the clinical outcomes of drug treatment. Thus, the HBV-infected organoid platform can potentially be used as an important pre-clinical, primary hepatocyte-screening platform for both the discovery of new candidate drugs targeting HBV transcription and replication and for assessing their potential toxicity.

When cultured under differentiating growth conditions, the expression of the HBV receptor NTCP increased significantly in liver organoids. Consistent with the observed higher NTCP levels, in vitro infection of differentiated organoids with HBV led to higher levels of infection and replication. HBV infection was dependent on NTCP, as treatment of liver organoids with the NTCP-targeting competitive HBV entry inhibitor myrcludex-B reduced infection. To obtain expanding long-term HBV-infected replicating cultures of human liver organoids using transgenic modification by lentiviral transduction, we generated liver organoids that exogenously expressed membrane-localized NTCP under expansion conditions. Surprisingly, the higher expression of NTCP did not result in more efficient infection of organoids, suggesting that although necessary, NTCP alone may not be sufficient for optimal infection (*Figure 3*). Consistent with our observations, HepG2 cell lines exogenously expressing NTCP, while susceptible to infection, are inefficiently infected with HBV and require high viral titers (*Iwamoto et al., 2014*; *Yan et al., 2012*). These observations are suggestive of the potential necessity for additional (co)receptors or downstream factors necessary for optimal infection in hepatocytes. Using lentiviral transduction, we also generated polyclonal liver organoids that contain integrated full-length HBV genome, which efficiently produced HBV under expansion conditions. This model, similar to HepG2.2.15 cells, although not a cccDNA-generating model of productive HBV infection, is a useful primary human liver model to investigate HBV transcription and HBV-directed drugs.

Recently, alternative liver organoid model systems derived from human (induced) pluripotent stem cells (*Kaneko et al., 2016*; *Nie et al., 2018*; *Sakurai et al., 2017*; *Xia et al., 2017*) as well as primary human hepatocyte co-culture systems (*March et al., 2015*; *Xiang et al., 2019*) have been developed, which can be used to study HBV infection. However, the human liver organoid platform offers two main advantages over these model systems. The first and most important is that HBV-infected patient-derived organoids resemble the diseased tissue of origin, as demonstrated by our gene expression studies (*Figure 5*). Second, the organoid platform allows the possibility of expansion and biobanking, ensuring the availability of sufficient samples for downstream analyses. Therefore, while primary human hepatocytes and iPSC models, similar to the ex vivo infection of hD human liver organoids we describe here, provide reliable ex vivo infection models for studies on HBV, the human liver organoid platform is uniquely suitable for long-term, patient-derived studies.

Early surveillance of the changes in gene expression or biomarkers that predict the occurrence of HCC is critical for enabling patients to receive timely and successful treatment. Paving the way for personalized therapy of HBV-infected patients, liver organoids can be seeded from infected patient liver resections, at different stages of the disease. Comparison of the gene expression profiles of

healthy donor to HBV-infected patient-derived liver organoids and further HDV-co-infected and acute liver failure patient sub-groups points to the extent to which organoids reflect the biology and represent the transcriptomic profiles of the primary non-tumor tissue of origin. Interestingly, although we found no sign of viral production in the non-tumor HBV-infected patient-derived liver organoid lines, we did observe portions of the HBV genome corresponding to parts of HBx integrated into these patient-derived organoids. Aside from the potential for insertional mutagenisis and disruption of host gene integration of HBx, a characterized oncogene may lead to aberrant host gene expression, providing a potential mechanism for iDonor organoids to display an early gene signature despite the absence of active HBV infection (*Liu et al., 2016*; *Figure 6A–B*). Thus, the observed HBV-infected patient-specific gene expression profile is likely a marker of HBV infection and possibly HBx expression. Our observations are consistent with the notion that the gene expression profile of the tumor microenvironment may serve as an important biomarker of HCC as bystander cells contain portions of the integrated HBV genome and are chronically exposed to released HBV proteins despite not having active viral replication. Indeed, our data indicate that HBV-infected non-tumor patient-derived liver organoids exhibit an early cancer gene profile, as shown by comparison to the TCGA-LIHC database. The identification of early aberrant gene regulatory networks and biomarkers that drive HBV-mediated HCC, detectable at an early non-tumor stage without phenotypic signs of tumorigenesis, allows for patient-specific surveillance of disease progression and provides a screening platform for candidate drugs that may guide personalized treatments by targeting early stages in HBV-mediated HCC.

## Materials and methods

### Key resources table

| Reagent type (species) or resource | Designation | Source or reference | Identifiers | Additional information |
|---|---|---|---|---|
| Cell line (*Homo sapiens*) | HepG2 | ATCC | HB-8065 RRID:CVCL_0027 | |
| Cell line (*Homo sapiens*) | HepG2.2.15 | CCTCC | CCTCC-GDC0141 RRID:CVCL_L855 | |
| Antibody | Hepatitis B Virus Core Antigen Antibody (1-5) (mouse monoclonal) | ThermoFisher | RRID:AB_1075044 | IF: (1:100) |
| Antibody | Anti-Hepatitis B Virus Core Antigen antibody (rabbit polyclonal) | Abcam | RRID:AB_10903422 | IF: (1:400) |
| Antibody | Hepatitis B Virus Surface Antibody (S 26) (mouse monoclonal) | ThermoFisher | RRID:AB_1075056 | IF: (1:200) |
| Antibody | Anti-ALB/Albumin Antibody (F-10) (mouse monoclonal) | Santa Cruz | sc-271605 RRID:AB_10647230 | IF: (1:50) |
| Antibody | HNF-4alpha (H-171) antibody (rabbit polyclonal) | Santa Cruz | sc-8987 RRID:AB_2116913 | IF: (1:50) |
| Antibody | Anti-SLC10A1 antibody produced in rabbit | Sigma-Aldrich | SAB2104761 RRID:AB_10668685 | IF: (1:200) |
| Antibody | Purified Anti-β-Catenin (mouse monoclonal) | BD Transduction Laboratories | 610154 RRID:AB_397555 | IF: (1:200) |
| Antibody | ANTI-FLAG M2 antibody (mouse monoclonal) | Sigma-Aldrich | F3165 RRID:AB_259529 | IF: (1:200) |
| Recombinant DNA reagent | pENTR/D-TOPO (plasmid backbone) | ThermoFisher | Invitrogen: K240020 | |
| Recombinant DNA reagent | pLenti6/V5-DEST Gateway Vector (plasmid backbone) | ThermoFisher | Invitrogen: V49610 | |

*Continued on next page*

Continued

| Reagent type (species) or resource | Designation | Source or reference | Identifiers | Additional information |
|---|---|---|---|---|
| Commercial assay or kit | Monolisa HBsAg ULTRA | Biorad | Biorad: 72346 | |
| Commercial assay or kit | Human Hepatitis B e Antigen (HBeAg) ELISA Kit | Cusabio | CSB-E13557h | |
| Commercial assay or kit | RealiaPrep RNA Cell Miniprep System | Promega | Promega: Z6012 | |
| Commercial assay or kit | DNase I | ThermoFisher | Invitrogen: 18047019 | |
| Commercial assay or kit | SuperScript II Reverse Transcriptase | ThermoFisher | Invitrogen: 18064022 | |
| Commercial assay or kit | GoTaq qPCR Master Mix | Promega | Promega: A6001 | |
| Commercial assay or kit | QIAamp MinElute Virus Spin Kit | Qiagen | Qiagen: 57704 | |
| Commercial assay or kit | DNeasy Blood and Tissue Kits | Qiagen | Qiagen: 69504 | |
| Commercial assay or kit | T5 Exonuclease | New England Biolabs | NEB: M0363 | |
| Commercial assay or kit | Plasmid-Safe ATP-Dependent DNase | Epicentre | E3101K | |
| Chemical compound, drug | Tenofovir disoproxil fumarate | Sigma-Aldrich | SML1794 | |
| Chemical compound, drug | Fialuridine | Cayman Chemical | 15867 | |
| Chemical compound, drug | Myrcludex-B | Creative Peptides | 10-101-139 | |
| Chemical compound, drug | Blasticidine S hydrochloride | Sigma-Aldrich | Sigma: 15205 | |
| Chemical compound, drug | Collagenase D | Sigma-Aldrich | Sigma: 11088858001 | |
| chemical compound, drug | TrypLE Express | ThermoFisher | Gibco:12604013 | |
| Chemical compound, drug | Cultrex Reduced Growth Factor Basement Membrane Extract, Type 2, Pathclear | R and D Systems | 3533-010-02 | |
| Chemical compound, drug | B-27 Supplement, minus vitamin A | ThermoFisher | Gibco: 12587010 | |
| Chemical compound, drug | N-2 Supplement | ThermoFisher | Gibco: 17502048 | |
| Chemical compound, drug | N-Acetyl-L-cysteine | Sigma-Aldrich | Sigma: A7250 | |
| Chemical compound, drug | Nicotinamide | Sigma-Aldrich | Sigma: N3376 | |
| Chemical compound, drug | Recombinant [Leu15]-Gastrin I human | Sigma-Aldrich | Sigma: G9145 | |
| Chemical compound, drug | Recombinant Human EGF | PeproTech | GMP100-15 | |
| Chemical compound, drug | Recombinant Human FGF-10 | PeproTech | 100–26 | |
| Chemical compound, drug | Recombinant Human HGF | PeproTech | 100–39H | |

*Continued*

| Reagent type (species) or resource | Designation | Source or reference | Identifiers | Additional information |
|---|---|---|---|---|
| Chemical compound, drug | Forskolin | Sigma-Aldrich | Sigma: F3917 | |
| Chemical compound, drug | Recombinant Human Noggin | PeproTech | 120–10C | |
| Chemical compound, drug | A 83–01 | Tocris | Tocris: 2939 | |
| Chemical compound, drug | ROCK Inhibitor (Y-27632) | Sigma-Aldrich | Sigma: SCM075 | |
| Chemical compound, drug | DAPT | Sigma-Aldrich | Sigma: D5942 | |
| Chemical compound, drug | Dexamethasone | Sigma-Aldrich | Sigma: D4902 | |
| Chemical compound, drug | Recombinant Human BMP-7 | PeproTech | 120–03P | |
| Chemical compound, drug | Recombinant Human FGF-19 | PeproTech | 100–32 | |
| Software, algorithm | Code used for RNA-Seq data analysis | *Moulos and Hatzis, 2015* | | The R package used for the RNA-Seq data analysis is available at https://www.bioconductor.org/packages/release/bioc/html/metaseqR.html |
| Software, algorithm | GraphPad Prism | https://graphpad.com | RRID:SCR_002798 | Version 5 |

## Liver tissue

The Medical Ethical Council of the Erasmus Medical Center approved the use of this material for research purposes (reference number: MEC-2014–060, Department of Surgery, Erasmus MC), and informed consent was provided from all patients. Biopsies from explanted HBV-infected livers and HBV-infected donor liver were fixed for 24 hr at room temperature (RT) in 4% formaldehyde solution (Klinipath) immediately after collection in the operating room. Fixed biopsies were processed according to the standard protocol to dehydrate and infiltrate with paraffin wax and subsequently embedded into paraffin blocks. Sections of 4 μm were cut using a microtome and mounted on glass microscope slides. After deparaffination according to the standard procedure, the tissue slides were stained with haematoxylin and eosin (Haemalum – Mayer's, VWR) according to the manufacturer's protocol, dehydrated, and mounted for microscopic analysis.

## Dissociation of human liver hepatocytes and seeding of organoids

Liver specimens from healthy donors and HBV-infected patients were dissociated and seeded to grow organoids using the method previously described by *Huch et al., 2015* with minor modifications. In brief, liver specimens (1–2 cm$^3$) were washed once with Dulbecco's Modified Eagle Medium (DMEM) (Sigma) supplemented with 1% fetal calf serum (FCS) and with 0.1% penicillin/ streptomycin (PS, Sigma), minced, and incubated at 37°C with the digestion solution (collagenase 2.5 mg/ml in Earle's Balanced Salt Solution [EBSS]). Incubation was performed for 30 min, further mincing and mixing the tissue every 10 min. To recover the cells, digestion solution was passed through a 70 μM strainer in a 50 ml tube (GreinerBio) and washed with 45 ml Advanced DMEM (Gibco) supplemented with 1% phosphatidylserine (PS), 10 mM 4-(2-hydroxyethyl)-1-piperazineetha-nesulfonic acid (HEPES) (Gibco), and 1% GlutaMax (Gibco), henceforth Ad+++. Partially digested tissue was recovered from the strainer and further incubated with TrypLE Express (Thermoscientific) for 15 min at 37°C. Cells obtained from the first and second digestion were pooled together and washed twice with Ad+++.

After the second centrifugation (200 x*g*, 5 min), cells were counted and 1–3 x 10⁶ cells were collected in 1 ml Trizol reagent (Sigma) for RNA isolation by following the manufacturer's instructions. To seed organoids, approximately 10,000–15,000 cells were mixed with a fixed amount of BME solution (ratio: 2/3 Basement Membrane Extract, type 2 (Pathclear), diluted with 1/3 Ad+++) and seeded in 25 µl drops in 48-well suspension plates (GreinerBio). After incubation at 37°C from 30 min, the BME solution had solidified and wells were filled with 250 µl of human liver organoid isolation medium consisting of Ad+++ supplemented with 1X B27 supplement without retinoic acid (Gibco), 1X N2 supplement (Gibco), 1.25 mM N-acetyl-L-cysteine (Sigma), 20% (vol/vol) Rspo-1 conditioned medium (*Huch et al., 2013*), 1.25% (vol/vol) Wnt3a conditioned medium (*Barker et al., 2010*), 10 mM nicotinamide (Sigma), 10 nM recombinant human (Leu15)-gastrin I (Sigma), 50 ng/ml recombinant human epidermal growth factor (EGF) (Peprotech), 100 ng/ml recombinant human fibroblast growth factor 10 (FGF10) (Peprotech), 25 ng/ml recombinant human hepatocyte growth factor (HGF) (Peprotech), 10 µM forskolin (Sigma), 5 µM A8301 (Tocris), 25 ng/ml Noggin (Peprotech), and 10 µM Y27632 Rho Kinase (ROCK) Inhibitor (Sigma).

## Expansion and maintenance of organoids in culture

After 1 week of seeding the organoids, isolation media were changed to human liver expansion media (EM; Ad+++ supplemented with 1X B27 supplement without retinoic acid (Gibco), 1X N2 supplement (Gibco), 1.25 mM N-acetyl-L-cysteine (Sigma), 20% (vol/vol) Rspo-1 conditioned medium, 1.25% (vol/vol) Wnt3a conditioned medium [*Barker et al., 2010*], 10 mM nicotinamide (Sigma), 10 nM recombinant human (Leu15)-gastrin I (Sigma), 50 ng/ml recombinant human EGF (Peprotech), 100 ng/ml recombinant human FGF10 (Peprotech), 25 ng/ml recombinant human HGF (Peprotech), 10 µM forskolin (Sigma), and 5 µM A8301 (Tocris)) (*Huch et al., 2015*).

EM was changed twice a week, and cultures were split every 7–10 days according to organoid density. For passaging (1:4-1:8, depending on growth rate of the culture), organoids were resuspended in 10 ml Ad+++, incubated in ice for 10 min, and collected by centrifugation (5 min at 200 x*g*). Subsequently, organoids were incubated for 1–2 min in TrypLE Express at RT and mechanically disrupted by pipetting. After a further wash in Ad+++, cells were resuspended in BME solution and seeded in 24- or 48-well suspension plates. After BME solution had solidified, wells were filled with 500 µl (24 wells) or 250 µl (48 wells) of human liver organoid expansion medium.

## Hepatic differentiation of liver organoids

Human liver organoid cultures derived from healthy and HBV-infected livers were seeded and cultured for 4 days in EM without Wnt3a conditioned medium supplemented with 25 ng/ml of Bone Morphogenetic Protein 7 (BMP7) (Peprotech). Hepatic differentiation was induced by culturing human liver organoids in DM (Ad+++ supplemented with 1X B27 supplement without retinoic acid, 1X N2 supplement, 1 mM N-acetylcysteine, 10 nM recombinant human [Leu15]-gastrin I, 50 ng/ml recombinant human EGF, 25 ng/ml recombinant human HGF, 0.5 µM A83-01, 10 µM N-[N-(3,5-Difluorophenacetyl)-L-alanyl]-S-phenylglycine t-butyl ester (DAPT) (Sigma), 3 µM dexamethasone (Sigma), 25 ng/ml BMP7, and 100 ng/ml recombinant human FGF19 (Peprotech)). DM was changed twice a week for 7 days before infection or 10 days before staining for albumin and HNF4α (*Huch et al., 2015*). The differentiated hepatocyte markers, which indicate liver cell properties, remain stably expressed in DM organoids and can be maintained for up to 2 months with intermittent culturing in expansion media. For subsequent downstream analysis, the following amounts of organoids were required (*Table 2*).

**Table 2.** Recommended scale of starting material.

| Procedure | Number of wells/condition | Number of organoids/well |
|---|---|---|
| DNA isolation | One well of a 24-well plate | ≈100–200 organoids/well |
| RNA isolation | One well of a 24-well plate | |
| cccDNA isolation | 20–24 wells of a 24-well plate | |
| Immunofluorescence staining | 5–10 wells of a 24-well plate | |

## Total RNA isolation and quantitative RT-qPCR

RNA extraction was performed starting from 1 to 2 wells of a 24-well plate. Organoids were collected either in 500 µl of cell lysis buffer and processed using RealiaPrep RNA Cell Miniprep System (Promega), according to the manufacturer's instructions, or in 1 ml of Trizol reagent (Sigma) and RNA was extracted according to the manufacturer's instructions. Purified RNA was treated with DNase I (Invitrogen) to remove potential HBV DNA contamination and, subsequently, cDNA synthesis was performed starting from 300 to 1000 ng of RNA using Superscript II Reverse Transcriptase (Invitrogen) kit following manufacturer's protocol for random primer cDNA synthesis. cDNA was diluted either 1:5 or 1:2.5 in nuclease-free water and 2 µl of the diluted product was used for real-time PCR with the following reagents: 5 µl of GoTaq qPCR Master Mix (Promega), 2 µl of nuclease-free water, and 1 µl of 10 mM primer mix. Amplification was performed on the CFX Connect Real-Time PCR Detection System thermocycler (BioRad) using the following thermal program starting with 3 min at 95°C, followed by 40 cycles of 95°C for 10 s and 60°C for 30 s. Specificity of the quantitative reverse transcription PCR (RT-qPCR) products was assessed by melting curve analysis. No RT control was routinely included in the procedure. Primers used for real-time qPCR are listed below.

GAPDH Fwd 5'-CAAGAAGGTGGTGAAGCAG-3'; Rev 5'-GCCAAATTCGTTGTCATACC-3'
KRT7 Fwd 5'-CTCCGGAATACCCGGAATGAG-3'; Rev 5'-ATCACAGAGATATTCACGGCTCC-3'
CYP3A4 Fwd 5'-TGTGCCTGAGAACACCAGAG-3'; Rev 5'-GTGGTGGAAATAGTCCCGTG-3'
NTCP Fwd 5'-GCTTTCTGCTGGGTTATGTTCTC-3'; Rev 5'-CATCCAGTCTCCATGCTGACA-3'
HNF4A Fwd 5'-CGTGCTGCTCCTAGGCAATGAC-3'; Rev 5'-ACGGACCTCCCAGCAGCATCT-3'
SOX9 Fwd 5'-GGAAGTCGGTGAAGAACGGG-3'; Rev 5'-TGTTGGAGATGACGTCGCTG-3'
ALBUMIN Fwd 5'-GCGACCATGCTTTTCAGCTC-3'; Rev 5'-GTTGCCTTGGGCTTGTGTTT-3'
LGR5 Fwd 5'-AGGTCTGGTGTGTTGCTGAGG-3'; Rev 5'-TGAAGACGCTGAGGTTGGAAGG-3'
LDLR Fwd 5'-GACGTGGCGTGAACATCTG-3'; Rev 5'-CTGGCAGGCAATGCTTTGG-3'
PCSK9 Fwd 5'-AGGGGAGGACATCATTGGTG-3'; Rev 5'-CAGGTTGGGGGTCAGTACC-3'
ADAMTS1 Fwd 5'-GGACAGGTGCAAGCTCATCTG-3'; Rev 5'-TCTACAACCTTGGGCTGAAAA-3'
CCNA1 Fwd 5'-GAGGTCCCGATGCTTGTCAG-3'; Rev 5'-GTTAGCAGCCCTAGCACTGTC-3'
CORO1a Fwd 5'-TGCACCCAGACACGATCTACA-3'; Rev 5'-TGTCACGGCAGGAGGTACAA-3'
PTHLH Fwd 5'-ATTTACGGCGACGATTCTTCC-3'; Rev 5'- GCTTGGAGTTAGGGGACACC-3'
STMN2 Fwd 5'-GCTCTCTCCGCTGCTGTAG-3'; Rev 5'-GGCCATTGCTGTTTTAGCCA-3'
STY5 Fwd 5'-AGACGCTGAACCCTCACTTTG-3'; Rev 5'-CGAAGTCGTACACCGCCAT-3'
Cyclophilin A Fwd 5'-TCATCTGCACTGCCAAGACTG-3'; Rev 5'-CATGCCTTCTTTCACTTTGCC-3'

Gene expression levels were calculated using the 2ΔCt method, whereas fold increase was calculated using the 2ΔΔCt method (*Schmittgen and Livak, 2008*). GAPDH and cyclophillin A were used as housekeeping controls as indicated.

## Detection of HBV RNA

cDNA obtained from reverse transcription of intracellular RNA (see previous section for details) was analyzed using a TaqMan-based qPCR assay. For each reaction, a 25-ul mixture was prepared containing 2.5 µl 10X buffer, 1.75 µl 25 mM $MgCl_2$, 1 µl 10 mM dNTPs, 1U of Platinum Taq, 0.125 µl of 100 µM forward (5'-GCAACTTTTTCACCTCTGCCTA-3') and reverse (5'-AGTAACTCCACAGTAGCTCCAAATT-3') primers, 0.075 µl of 50 µM probe (FAM-TTCAAGCCTCCAAGCTGTGCCTTGGGTGGC-BHQ1), and 4 µl (cDNA) of template. Beta-2-microglobulin was used as a housekeeping control for expression analysis of cDNA samples (B2M Fwd 5'-AGCGTACTCCAAAGATTCAGGTT-3', B2M Rev 5'-ATGATGCTGCTTACATGTCTCGAT-3', B2M probe FAM-TCCATCCGACATTGAAGTTGACTTACTG-BHQ1).

Detection of the 3.5-kb RNA transcript and total HBV RNA was performed using a previously published nested PCR protocol (*Wong et al., 2011*) followed by agarose gel electrophoresis with cDNA samples as templates for the nested PCR. Primers used for the nested PCR are listed below.

3.5 kb_Ext_Fwd 5'-CGCGTCGCAGAAGATCTCA-3'; Rev 5'-CGAGTCTAGACTCTGCGGTA-3'

3.5 kb Fwd 5′-GTATYCCTTGGACTCATAAGGTGGG-3′; Rev 5′-CTTGTTCCCAAGAATATGG TGACCC-3′
Total HBV_Ext_Fwd 5′-CGCCAACTTACAAGGCCTTTCT-3′; Rev 5′-GTCGGTCGTTGACA TTGCAGAGAG-3′
Total HBV_Fwd 5′-GCTAGGCTGTGCTGCCAACTG-3′; Rev 5′-GGGGAGTCCGCGTAAAGA-GAGG-3′

## Detection of HBsAg and HBeAg

50 µl of the organoid supernatant was collected after 3–6 days post infection and used for the detection of HBsAg and HBeAg. Monolisa HBs Ag ULTRA (BioRad) and Human Hepatitis B e Antigen (HBeAg) ELISA Kit (Cusabio) were used for the detection of HBsAg and HBeAg, respectively, from the supernatant of the infected organoids following manufacturer's instructions. Absorbance values at 450 nm for the samples were normalized by the negative control provided by the manufacturers and depicted as signal over control (S/CO).

## Immunofluorescence and image analysis

Human liver organoids were collected and washed three times with cold Ad+++ to remove BME, then fixed with 4% paraformaldehyde for 30 min in ice, and permeabilized using 0.3% (HBcAg, HNF4α, and NTCP) or 1% (albumin) Triton X-100 (Sigma) in phosphate-buffered saline (PBS) for 30 min at RT. For HBsAg staining, cells were fixed and permeabilized in 100% acetone. Specimens were incubated for 2 hr at RT in PBS plus 10% bovine serum albumin (BSA) (Roche) and 0.5% FCS (HBsAg) or PBS plus 0.5% FCS, 0.3% triton, 1% BSA 1% DMSO (albumin, HBcAg, HNF4α, and SLC10A1). Following blocking, human liver organoids were incubated overnight with primary antibodies (mouse anti-HBcAg, mouse anti- HBsAg (ThermoScientific), mouse anti-HBcAg (Abcam)), mouse anti-albumin, rabbit anti-HNF4α (Santa Cruz Biotechnology), and rabbit anti-SLC10A1 (NTCP, Sigma) diluted in PBS + 10% blocking buffer. After extensive washing, human liver organoids were stained with appropriate Alexa Fluor dye-conjugated secondary antibodies (Life Technologies). Nuclei were stained with Hoechst33342 (Molecular Probes). Immunofluorescence images were acquired using a confocal microscope (Leica, SP5). Images were analyzed and processed using Leica LAS AF Lite software (Leica SP5 confocal). All phase-contrast pictures were acquired using a Leica DMIL microscope and a DFC420C camera.

## Production of HBV virus and HBV infection

HepG2.2.15 cells, a HepG2-derived cell line stably transfected with full-length HBV (kindly provided by Prof. Bart Haagmans, Erasmus MC), were cultured in DMEM medium (Gibco) supplemented with 10% fetal bovine serum (Gibco) and 1% penicillin/streptomycin. For virus production, $3 \times 10^6$ cells were plated in collagen-coated 10 cm plates, cultured in supplemented DMEM until confluency and subsequently in Ad+++ for 4 days. The supernatant of HepG2.2.15 cells was then collected, filtered, and concentrated using the PEG Virus Precipitation Kit (Abcam) following the manufacturer's instructions. Precipitated virus was aliquoted and stored at −80°C until use. Human serum was obtained from residual samples from HBV-infected individuals attending Erasmus MC for routine clinical activity. As a negative control, an aliquot of the virus equivalent to the inoculum was inactivated by incubation at 100°C for 30 min. Human liver organoids were resuspended using either active virus or heat-inactivated control at a multiplicity of infection (MOI) of $1–10 \times 10^3$ copies HBV DNA/organoid, transferred to 24-well plates, and centrifuged for 1 hr at 600 xg. Following spinoculation, plates were incubated at 37°C for 5 hr and then seeded in BME following the culturing protocol. After BME solution has solidified, liver organoids were maintained in EM for 16 hr, washed four times with Ad+ ++, and cultured in EM or DM as indicated. To check the NTCP dependency for infection, organoids were treated before infection for 2 hr with 10 µM myrcludex-B followed by the continuous presence of 10 µM myrcludex-B throughout the whole infection procedure. Myrcludex-B treatment was then continued till 6 days post infection.

## Isolation of HBV DNA

To isolate HBV DNA from the supernatants of HepG2.2.15 cells and organoid cultures infected with either infectious HepG2.2.15-produced virus or HI virus, the QIAamp MinElute Virus Spin Kit was used following the manufacturer's instructions.

To isolate intracellular HBV DNA from hD organoid cultures infected with either infectious HepG2.2.15-produced virus or HI virus, organoids were harvested after 6 days post infection, and DNeasy Blood and Tissue kit by Qiagen was used following the manufacturer's instructions to isolate total HBV DNA. Levels of cccDNA produced in HBV-infected hD organoid cultures were quantified either after DNA isolation (DNeasy Blood and Tissue kit by Qiagen) followed by T5 exonuclease digestion (New England Biolabs) or by a previously described alkali lysis plasmid DNA-isolation protocol with a minor modification (*Yang et al., 1996*) followed by digestion with plasmid-safe DNase (Epicentre, E3101K, Madison, WI). Briefly, hD liver organoids, infected with HepG2.2.15-produced or HI virus, were collected 6–8 days post infection, incubated on ice for 10–20 min, and washed with Ad+++ to remove the BME. For DNeasy Blood and Tissue kit, washed organoids were used for isolation of the total genomic DNA following the manufacturer's protocol. Isolated DNA was treated with T5 exonuclease to remove chromosomal DNA and nicked or linear HBV DNA. DNA was digested according to the manufacturer's instructions and T5 exonuclease was inactivated by adding at least 11 mM of ethylenediaminetetraacetic acid (EDTA). The reaction volume was purified using QIAquick PCR Purification Kit by Qiagen.

For the alkali lysis plasmid DNA-isolation protocol, organoids were treated with TrypLE Express and incubated for 30–50 s at RT. Organoids were washed with PBS, collected by centrifugation (5 min at 1000 rpm), and resuspended with 800 μl of ice-cold cell lysis buffer (1 mM EDTA (pH 8.0), 5 mM Tris HCl (pH 7.5), and 0.05% Nonidet P-40). After 10 min of incubation on ice, an equal volume of alkali lysis buffer (0.1 M NaOH, 6% sodium dodecyl sulfate (SDS)) was added and the solution was incubated for 30 min at 37°C. DNA was neutralized by adding 3 M potassium acetate (pH 5.0) to a final concentration of 0.6 M and centrifuged for 5 min at 12,000 rpm. The supernatant was extracted two times with phenol followed by extraction with butanol:isopropanol (7:3) for removal of any residual phenol. Subsequently, the DNA was precipitated with 1 ml 100% ethanol, 400 μl 7.5 M ammonium acetate, and 1 μl 20 mg/ml glycogen overnight at −80°C. Next day, the cccDNA sample was spun down for 30 min at 4°C, 14,000 rpm, and washed with 70% ethanol. After spinning the samples for 15 min at 4°C, 14,000 rpm, the pellet was resuspended in 50 μl of nuclease-free water. To remove the chromosomal DNA or any linear HBV DNA, 25 μl of the isolated DNA was digested with plasmid-safe DNase according to the manufacturer's protocol and the samples were digested for 1 hr at 37°C followed by 30 min of heat inactivation at 70°C. The digested samples were purified once with phenol:chloroform:isoamylalcohol (Sigma-Aldrich) followed by treatment with chloroform: isoamylalcohol (24:1), cccDNA was precipitated with 20 ul 3 M NaAC (pH 5.2), 1 ul glycogen (20 mg/ml), 1 ml 100% ethanol and washed as described above, and the final DNA pellet was resuspended in 30 μl of nuclease-free water.

## Detection of HBV DNA

HBV DNA extracted from culture supernatant and intracellular HBV DNA were analyzed in duplicate using a TaqMan-based qPCR assay. For each reaction, a 25-ul mixture was prepared containing 2.5 μl 10X buffer, 1.75 μl 25 mM MgCl$_2$, 1 μl 10 mM dNTPs, 1U of Platinum Taq, 0.125 μl of 100 μM forward (5'-GCAACTTTTTCACCTCTGCCTA-3') and reverse (5'-AGTAACTCCACAGTAGCTCCAAATT-3') primers, 0.075 μl of 50 μM probe (FAM-TTCAAGCCTCCAAGCTGTGCCTTGGGTGGC-BHQ1), and 7.5 μl (DNA) of template. Each PCR included a standard curve made of dilutions of a plasmid containing the full-length HBV genome ranging from 4 to 4 x $10^5$ copies of plasmid. Amplification was performed on the CFX Connect Real-Time PCR Detection System thermocycler (BioRad) using the following thermal program starting with 10 min at 95°C, followed by 45 cycles of 95°C for 10 s, 60°C for 30 s, and 72°C for 10 s.

For quantification of cccDNA, a TaqMan-based qPCR was performed, whereby for each reaction, a 20-μl reaction mix was prepared containing 4.2 μl of the cccDNA template, 10 μl LightCycler480 Probes Master (Roche), 1 μM primer mix (Fwd 5'-GTCTGTGCCTTCTCATCTGC-3'; Rev 5'-AGTAAC TCCACAGTAGCTCCAAATT-3'), 0.2 μM probe (FAM-TTCAAGCCTCCAAGCTGTGCCTTGGG TGGC-BHQ1), and 4% dimethyl sulfoxide (DMSO). qPCR was carried out using a previously

published protocol: 95℃ for 10 min, followed by 50 cycles of 95℃ for 15 s and 61℃ for 1 min (*Winer et al., 2017*). Lenti-HBV plasmid construct containing 1.3mer HBV genome (*Figure 3—figure supplement 2C*) was used as a positive control for the qPCR.

### Detection of integrated HBV DNA in HBV-infected patient-derived liver organoid genome

Intracellular DNA, isolated from iPs and iDs using DNeasy Blood and Tissue kit by Qiagen, was used to detect the region of HBV DNA integrated in the organoid genome. To amplify the integrated region, PCR containing ~500–800 ng of DNA template, 2X GoTaq qPCR Master Mix (Promega) to a final concentration of 1X, and primer mix to a final concentration of 500 nM was prepared. Amplification was performed on the CFX Connect Real-Time PCR Detection System thermocycler (BioRad) using the following thermal program starting with 3 min at 95℃, followed by 45 cycles of 95℃ for 10 s and 58℃ for 30 s. Primers used for PCR are listed below:

> PolA Fwd 5'-GCTTTCACTTTCTCGCCAAC-3'; Rev 5'-GTTGCGTCAGCAAACACTTG-3'
> PolB Fwd 5'-CGATCCATACTGCGGAACTC-3'; Rev 5'-TATTTGCGGGGAGAGGACAAC-3'
> XA Fwd 5'-AATCCTGCGGACGACCCT-3'; Rev 5'-CAGAGGTGAAGCGAAGTGC-3'
> XB Fwd 5'-TGCACTTCGCTTCACCTCTG-3'; Rev 5'-CTCAAGGTCGGTCGTTGACA-3'
> XC Fwd 5'-GGCATACTTCAAAGACTGTTTGTTT-3'; Rev 5'-CGCAGACCAATTTATGCCTAC-3'
> Core Fwd 5'-GCAACTTTTTCACCTCTGCCTA-3'; Rev 5'-AGTAACTCCACAGTAGCTCCAAA TT-3'
> Albumin Fwd 5'-TGCATGAGAAAACGCCAGTAA-3'; Rev 5'-ATGGTCGCCTGTTCACCAA-3'

PCR-amplified DNA fragments corresponding to different regions of HBV Core, X, and Pol as indicated were visualized by running the fragments on 1.5% agarose gels. Gel images were acquired using Uvidoc HD2 (Uvitec Cambridge) gel documentation system.

### Infection with HBV generated from organoids

The supernatant produced by infected organoids at 4–8 days post infection was collected and concentrated using Amicon Ultra-15 100K (Milipore). Human liver organoids were resuspended using either concentrated HBV virus or HI control. After an hour of spinoculation at 32℃, 600 xg, the plate was incubated at 37℃ overnight and then washed and seeded in BME following the HBV infection and culturing protocol.

### Cell culture

HepG2 cells were obtained from ATCC and HepG2.2.15 cells were obtained from CCTCC. The identity of the cells was authenticated by the manufacturers by short tandem repeats (STR) profiling and cell lines were routinely tested for mycoplasma. HepG2 and HepG2.2.15 cells were cultured in DMEM medium (Gibco) supplemented with 10% fetal bovine serum (Gibco) and 1% penicillin/streptomycin and incubated in 5% $CO_2$ at 37℃.

### Viability assays

HepG2 cells with a cell density of $2 \times 10^4$ cells/ml were seeded in 24-well plates with DMEM high-glucose media supplemented with fetal bovine serum (10% v/v) and penicillin/streptomycin (1% v/v) and incubated in 5% $CO_2$ at 37℃ for overnight. Human liver organoids seeded from healthy donor or HBV-infected patient livers were split in the ratio 1:10, seeded in 20 µl BME 3D culture in 48-well plates, and differentiated as described above. 10 days post differentiation for human liver organoids and 24 hr post seeding of the HepG2 cells, they were treated with different concentrations of Fialuridine (Cayman, 15867–1) or Tenofovir disproxil fumarate (Sigma, SML1794) (1–20 µM) or the vehicle control. At different time points (2–10 days) post treatment, organoid viability was measured using the alamarBlue viability assay (alamarBlue Invitrogen DAL1025, 1:10 in DM) according to the manufacturer's instructions. Briefly, treatment DM medium was removed from wells and 10% alamarBlue with DM was added to each well and incubated for 4 hr (organoids) or 2 hr (HepG2 cells) at 37℃ before absorbance readings were taken at 570 and 600 nm. The results were normalized to control vehicle-treated differentiated organoids. The lower limit of quantitation of the assay was determined by values obtained from BME without organoids and is represented by dotted lines in

figure panels. Each treatment condition was repeated at least three times and readings were done in duplicate. Cell imaging was performed after each alamarBlue assay.

## 3' mRNA sequencing

For each RNA preparation, two wells of organoids in expansion phase were collected in Trizol reagent (Sigma) 4–5 days after splitting. RNA was extracted according to the manufacturer's instruction and resuspended in 30 µl of nuclease-free water. Total RNA was quantitated (ND1000 Spectrophotometer – PEQLAB). Samples were diluted accordingly to a mean concentration of approximately 100–150 ng/µl and their quality assessed on a Bioanalyzer (Agilent Technologies) using the Agilent RNA 6000 Nano Kit reagents and protocol (Agilent Technologies). RNA samples were processed for library preparation using the 3' mRNA-Seq Library Prep Kit Protocol for Ion Torrent (QuantSeq-LEXOGEN, Vienna, Austria), according to the manufacturer's instructions. Briefly, up to 500 ng from each RNA sample was used for first strand synthesis. The RNA was subsequently removed and second-strand synthesis was initiated by a random primer, containing Ion Torrent compatible linker sequences and appropriate in-line barcodes. Second-strand synthesis was followed by magnetic bead-based purification and the resulting library was PCR-amplified for 14 cycles and re-purified. Library quality and quantity was assessed on a Bioanalyzer using the DNA High Sensitivity Kit reagents and protocol (Agilent Technologies). The quantified libraries were pooled together at a final concentration of 100 pM. The pools were templated and enriched on an Ion Proton One Touch system. Templating was performed using the Ion PI Hi-Q OT2 200 Kit (Thermo Fisher Scientific), followed by sequencing using the Ion PI Hi-Q Sequencing 200 Kit on Ion Proton PI V2 chips (Thermo Fisher Scientific), an Ion Proton System, according to the manufacturer's instructions.

## Short-read mapping

The Quant-Seq FASTQ files obtained from Ion Proton sequencing were mapped on the UCSC hg19 reference genome using a two-phase mapping procedure. Firstly, the short reads were mapped using tophat2 (*Kim et al., 2013*), with the following non-default settings: `–read-mismatches 3 – read-gap-length 3 –read-edit-dist 3 –no-novel-juncs`, other settings at default. Additional transcript annotation data for the hg19 genome from Illumina iGenomes (http://cufflinks.cbcb. umd.edu/igenomes.html) were also provided to tophat2 for guidance. Next, the reads that remained unmapped were converted back to FASTQ files using bam2fastq from the BEDTools (*Quinlan and Hall, 2010*) suite and submitted to a second round of mapping using Bowtie2 (*Langmead and Salzberg, 2012*) against the hg19 genome with the `–local` and `–very-sensitive-local` switches turned on. All resulting BAM files were visualized in the UCSC Genome Browser using BEDTools and tools provided by the UCSC Genome Browser toolkit.

## Statistical analysis of Quant-Seq data

The resulting Quant-Seq BAM files were analyzed with the Bioconductor package metaseqR (*Moulos and Hatzis, 2015*), which has built-in support for Quant-Seq data. Briefly, the raw BAM files, one for each organoid sample, were summarized to a 3' UTR read counts table from Ensembl longest (dominant) transcripts (version 90). The original 3' UTR regions were extended 500 bp upstream and downstream to accommodate the variable read length of Ion Proton reads. In the resulting read counts table, each row represented one 3' UTR region, each column one Quant-Seq sample, and each cell the corresponding read counts associated with each row and column. The final 3' UTR read counts table was normalized for inherent systematic or experimental biases using the Bioconductor package DESeq (*Anders et al., 2012*) after removing areas that had zero counts over all the Quant-Seq samples. Prior to the statistical testing procedure, the 3' UTR read counts were filtered for possible artifacts that could affect the subsequent statistical testing procedures. 3' UTR areas presenting any of the following were excluded from further analysis: (i) 3' UTR areas corresponding to genes smaller than 500 bp, (ii) 3' UTRs with read counts below the median read counts of the total normalized count distribution (similar expression thresholds [e.g., the median of the count distribution] have been previously used in the literature [*Mokry et al., 2012*], where the authors use the median reads per kilobase of transcript, per million mapped reads (RPKM) value instead of normalized counts), and (iii) 3' UTR areas corresponding to genes with the following Ensembl biotypes: rRNA, TR_V_pseudogene, TR_J_pseudogene, IG_C_pseudogene,

IG_J_pseudogene, IG_V pseudogene. The remaining 3' UTR counts table after filter application was subjected to differential expression analysis for the appropriate contrasts using the PANDORA algorithm implemented in metaseqR. 3' UTR areas (and their corresponding genes) presenting a PANDORA p-value less than 0.05 and a fold change (for each contrast) greater than 1 or less than −1 in log2 scale were considered as differentially expressed.

## Clustering analysis

Hierarchical clustering was performed using the Euclidean distance and complete linkage for the construction of the dendrograms. The expression values used to generate the heatmap were DESeq-normalized read counts in log2 scale. Multidimensional scaling was performed using the Spearman correlation distance metric on the gene expression matrix using DESeq-normalized read counts in log2 scale. All calculations and visualizations were performed using facilities from the R language. All analysis scripts and logs of the analysis pipelines are available upon request.

## Gene ontology and pathway analysis

GO enrichment and biochemical pathway analysis was performed using GeneCodis (*Tabas-Madrid et al., 2012*). For the GeneCodis GO and pathway analysis, the 361 iP 'signature' genes were used.

## Generation of the lentiviral vectors and transduction of liver organoids

A gene block fragment encoding human NTCP was designed based on the reference sequence retrieved from NCBI nucleotide database (NM_003049.3). A 3X Flag PCR fragment including NTCP coding sequence was amplified using the NTCP_FWD (CACCATGGATTACAAGGATGACGACGA TAAGGATTACAAGGATGACGACGATAAGGATTACAAGGATGACGACGATAAGATGGAGGCCCA-CAACGCGTCTgcccca) and NTCP_REV (TTACTAGGCTGTGCAAGGGGAGCA) primers and cloned in the pENTR/D-TOPO entry vector (Invitrogen) following the manufacturer's instructions.

A pENTR/D-TOPO vector harboring a competent full-length copy of HBV (1.3 wt genomes) was generated using a Gibson Assembly protocol. Two fragments encompassing 1.3 times the wt HBV genome were purified using the QIAquick PCR Purification Kit (Qiagen) after PCR using the following primer pairs: VectorFwd (5'-caaaaaagcaggctccgcgggccgcccccttcacGGACGACCCTTCTCGGGG-3')/MiddleRev (5'-gagaagtccaccacgAGTCTAGACTCTGCGGTATTGTGAG-3') and MiddleFwd (5'-cgcagagtctagactCGTGGTGGACTTCTCTCAATTTTC-3')/VectorRev (5'-tgccaactttgtacaa-gaaagctgggtcggAGGGGCATTTGGTGGTCTATAAG-3'). The HepG2.2.15 DNA was used as the template for the PCR: an empty pENTR/D-TOPO was cut with NotI and BssHI and gel purified using QIAquick Gel Extraction Kit (Qiagen). After purification, the entry vector and the PCR fragments were assembled using the Gibson Assembly Master Mix (New England BioLabs), following the manufacturer's instructions.

To generate lentiviral constructs, the fragments cloned into entry vectors were transferred to a lentiviral expression destination vector (pLenti6/V5-DEST Gateway Vector, Invitrogen), using the Gateway technology (Invitrogen). The full-length HBV pLenti6/V5-DEST vector was further modified in order to remove the CMV promoter. The vector was digested with two restriction enzymes flanking the CMV promoter, ClaI and PstI (New England Biolabs), and purified from gel using QIAquick Gel Extraction Kit. After filling end gaps using Klenow polymerase (New England Biolabs), plasmids were ligated and transformed in One Shot Stbl3 chemically competent *Escherichia coli* (Invitrogen). All HBV and NTCP lentiviral vectors were sequenced to verify the vector structure and integrity of open reading frames.

The lentiviral constructs were generated using the ViraPower Kit (Invitrogen) and 293FT cells, following the manufacturer's protocol. Briefly, 1 day prior to transfection, $3 \times 10^5$ cells were plated in 10 cm dishes in DMEM + 10% FCS. The following day, DMEM was replaced with 5 ml of Opti MEM I (Invitrogen) and cells were transfected with 9 µg of the ViraPower Packaging Mix and 3 µg of the pLenti6/V5-DEST Gateway Vector using Lipofectamine (Invitrogen). The day after, the transfection medium was changed to DMEM + 10% FCS. Cell supernatant containing the lentiviral particles was collected 60 and 72 hr after transfection, filtered with a 0.42 µm filter, aliquoted, and stored at −80° C.

Early-passage (passage 0–3) human liver organoids were collected in cold Ad+++ and incubated for 10 min in ice to remove BME. After centrifugation (5 min at 200 x*g*), organoids were resuspended in TrypLE Express and incubated until single cells were >80%. Cells were collected by centrifugation (5 min at 200 x*g*), resuspended in 1 ml of lentiviral harvest, and divided over four wells of a 48-well plate. Plates were centrifugated for 1 hr at 600 x*g* and then incubated for 5 hr at 37°C. Afterwards, cells were collected by centrifugation (5 min at 200 x*g*), resuspended in BME solution, and plated. After BME has solidified, wells were filled with EM supplemented with 25 ng/ml Noggin and 10 μM Y27632. 4 days after infection, the medium was changed to EM supplemented with 25 ng/ml Noggin and 10 μM Rho Kinase (ROCK) Inhibitor Y27632 and 5 ug/ml blasticidin. Organoids were kept under selection for 7 days and then the medium was changed to regular EM. Once selected organoids have recovered, cultures were split to remove dead cells and cultured according to the regular protocol. Parallel infections were performed using HepG2 cells and the same lentiviral constructs in order to generate control cell lines.

To determine NTCP functionality following transduction with NTCP lentiviral vector, parental and selected NTCP-transduced organoids were cultured in EM or EM supplemented with different concentrations of aruvastatin and rosuvastatin for 12 hr. Organoids from two wells of a 24-well plate were collected in 500 μl of cell lysis buffer and processed using RealiaPrep RNA Cell Miniprep System (Promega), according to the manufacturer's instructions.

To determine HBV production following transduction with the full-length HBV lentiviral vector, the supernatant and cells from transduced cultures were collected at different time points after completion of blasticidin selection. Presence of viral DNA in the supernatant and cellular-associated viral RNA was assessed using the real-time protocol detailed as mentioned before in detection of HBV DNA and RNA section. Presence of HBsAg in organoid supernatant was assessed using the Mona-Lisa Kit (Promega) according to the manufacturer's instructions.

## Statistical analysis

The data were first analyzed by analysis of variance (ANOVA), and then each pair was compared through Dunnett's multiple comparisons test. A value of $p < 0.05$ was considered statistically significant. Data are shown as mean ± SD of at least three replicate treatments; *$p < 0.05$; **$p < 0.01$. The data were analyzed and graphs were depicted by GraphPad Prism software 5 (GraphPad Software, La Jolla, CA, USA). Obtained data from drug screening of infected organoids have two to eight technical replicates for each donor. Thus, the obtained data from each donor were analyzed by ANOVA separately or as stated in the figure legends.

## Acknowledgements

We would like to thank Vaggelis Harokopos of the Fleming Genomics Facility for NGS and Karien Hamer for technical support. TM received funding from the European Research Council (ERC) under the European Union's Seventh Framework Programme (FP/2007–2013)/ERC STG 337116 Trxn-PURGE, Dutch AIDS Fonds grants 2014021 and 2016014, and Erasmus MC mRACE research grant.

## Additional information

### Funding

| Funder | Grant reference number | Author |
| --- | --- | --- |
| European Research Council | STG 337116 | Tokameh Mahmoudi |
| Aids Fonds | 201614 | Tokameh Mahmoudi |
| Bristol-Myers Squibb | AI424-543 | Elisa De Crignis |

The funders had no role in study design, data collection and interpretation, or the decision to submit the work for publication.

## Author contributions

Elisa De Crignis, Conceptualization, Resources, Formal analysis, Supervision, Funding acquisition, Validation, Investigation, Visualization, Methodology, Writing - original draft, Project administration, Writing - review and editing; Tanvir Hossain, Data curation, Formal analysis, Validation, Investigation, Methodology, Writing - review and editing; Shahla Romal, Conceptualization, Investigation, Visualization; Fabrizia Carofiglio, Conceptualization, Data curation, Supervision, Investigation, Visualization, Methodology, Writing - original draft; Panagiotis Moulos, Resources, Data curation, Software, Writing - original draft; Mir Mubashir Khalid, Formal analysis, Investigation, Methodology; Shringar Rao, Visualization, Methodology, Writing - review and editing; Ameneh Bazrafshan, Investigation, Visualization, Methodology; Monique MA Verstegen, Investigation, Visualization; Farzin Pourfarzad, Conceptualization, Investigation, Methodology; Christina Koutsothanassis, Investigation, Methodology; Helmuth Gehart, Meritxell Huch, Sylvia F Boj, Conceptualization, Methodology; Tsung Wai Kan, Investigation; Robert-Jan Palstra, Conceptualization, Software, Formal analysis; Charles Boucher, Conceptualization, Investigation, Writing - original draft; Jan NM IJzermans, Resources, Supervision, Investigation; Robert Vries, Conceptualization, Formal analysis, Investigation, Methodology, Writing - original draft; Hans Clevers, Conceptualization, Supervision, Methodology; Luc JW van der Laan, Conceptualization, Supervision, Investigation, Writing - original draft; Pantelis Hatzis, Conceptualization, Resources, Data curation, Software, Formal analysis, Supervision, Investigation, Writing - original draft, Writing - review and editing; Tokameh Mahmoudi, Conceptualization, Resources, Data curation, Software, Formal analysis, Supervision, Funding acquisition, Validation, Investigation, Visualization, Methodology, Writing - original draft, Project administration, Writing - review and editing

## Author ORCIDs

Tanvir Hossain (iD) https://orcid.org/0000-0002-4765-6600
Shahla Romal (iD) https://orcid.org/0000-0002-6404-0015
Mir Mubashir Khalid (iD) http://orcid.org/0000-0002-1862-6795
Tokameh Mahmoudi (iD) https://orcid.org/0000-0002-2060-9353

## Ethics

Human subjects: The Medical Ethical Council of the Erasmus Medical Center approved the use of this material for research purposes (reference number: MEC-2014-060, Department of Surgery, Erasmus MC), and informed consent was provided from all patients.

## Decision letter and Author response

Decision letter https://doi.org/10.7554/eLife.60747.sa1
Author response https://doi.org/10.7554/eLife.60747.sa2

# Additional files

## Supplementary files

- Supplementary file 1. Gene expression signature.
- Supplementary file 2. GO analysis.
- Transparent reporting form

## Data availability

Sequencing data that support the findings of this study have been deposited in GEO with the accession code GSE 126798.

The following dataset was generated:

| Author(s) | Year | Dataset title | Dataset URL | Database and Identifier |
|---|---|---|---|---|
| Mahmoudi T, De Crignis E, Moulos P | 2020 | Human liver organoids; a patient-derived primary model for HBV Infection, Replication | https://www.ncbi.nlm.nih.gov/geo/query/acc.cgi?acc=GSE126798 | NCBI Gene Expression Omnibus, GSE126798 |

and Related Hepatocellular
Carcinoma

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
