## [Decision Letter]

Thank you for submitting your article "Human liver organoids are a patient-derived primary model for HBV Infection and Related Hepatocellular Carcinoma" for consideration by *eLife*. Your article has been reviewed by 3 peer reviewers, including Melanie Ott as the Reviewing Editor and Reviewer #1, and the evaluation has been overseen by Didier Stainier as the Senior Editor. The following individual involved in review of your submission has agreed to reveal their identity: Aleem Siddiqui (Reviewer #3).

The reviewers have discussed the reviews with one another and the Reviewing Editor has drafted this decision to help you prepare a revised submission.

Summary:

The authors previously reported the human liver organoid system using human liver biopsy samples. In this manuscript, they develop human liver organoids as a model system for hepatitis B virus (HBV) infection and related hepatocellular carcinoma (HCC) and as a tool to screen for antiviral therapeutics. The authors demonstrate the ability to generate liver organoids from healthy as well as chronic HBV patients, and these organoids in the EM state retain stem cell markers and in DM upregulate hepatocyte markers as expected. In the expansion media, organoid hepatocytes expressed low levels of NTCP. Levels of NTCP expression was higher in the differentiation medium, but ultimately the authors established NTCP-overexpressed human liver organoids using lentiviral transduction. These were used for de novo infections with HBV and overall for drug-induced toxicity studies. The authors also performed bulk RNAseq analysis of organoids derived from HBV+ from patients (without de novo infection) to identify an "early signature" gene set that can serve as a biomarker for HBV infection and related HCC.

Essential revisions:

All three reviewers recognize the novelty and potential impact of the described liver organoid system on HBV research. Revisions are outlined to strengthen (1) the HBV virology; (2) the conclusions from the RNA-Seq data; (3) information regarding the organoid system.

1) The absolute numbers of cccDNA and intracellular HBV DNA, as well as their ratios, varied significantly among the liver organoids from different donors, which could be due to the batch-to-batch variation of liver organoids or the variations of the detection method. In these regards, the authors need to further characterize the viral nucleic acids (cccDNA, RNA, core DNA) in HBV-infected liver organoids more qualitatively and quantitatively.To unambiguously show successful infection of the organoids, detection of cccDNA by Southern blotting is the gold standard and should be shown. Alternatively, a northern blot of HBV RNA or Southern blot of HBV core DNA with and without Nuc treatment can be acceptable.

2) It would be critical to include organoids from cirrhotic donors who have not been previously infected in the analysis. Comparing infected versus uninfected "healthy" individuals makes interpreting the RNAseq data (Figure 4F) ambiguous as it remains unclear what weight the "cirrhotic" gene expression signature carries in the interpretation of results versus HBV infection. Also, were input samples analyzed, and are these transcriptome differences consistent with the liver samples before organoid formation? Several genes related to early HCC were up-regulated in HBV patient-derived liver organoids systems. However, the authors did not verify these genes and their expression levels in the HBV-associated HCC patients and HBV infected human liver organoids systems.

3) The authors have not stated how long NTCP-overexpressed human liver organoids and HBV patient-derived liver organoids systems can be kept in culture while maintaining liver cell properties. The authors should confirm these. Could the authors explain more on why HBV-positive liver organoids were not obtained from HBV patient livers? Have the authors detected HBV integration in the established organoids derived from HBV patients?

[Editors' note: further revisions were suggested prior to acceptance, as described below.]

Thank you for resubmitting your work entitled "Application of human liver organoids as a patient-derived primary model for HBV Infection and Related Hepatocellular Carcinoma" for further consideration by *eLife*. Your revised article has been evaluated by Didier Stainier as the Senior Editor, and a Reviewing Editor.

The manuscript has been improved but there are some remaining issues that need to be addressed, as outlined below:

The authors add new data to address some of the areas highlighted in the previous peer review including critically important RNA-seq analysis of non-cirrhotic donors and analysis of integration events in iP organoids. The authors also address several points regarding HBV virology, including elimination of the CMV-driven 1.3xHBV construct data.

A critical issue that the authors need to address is the hierarchical clustering analysis in Figure 6. The authors provide two analyses (Figure 6A and 6C) which share the iP organoid lines but not the organoids from donors that cleared infection (iD-B) and donors that had NASH (CN). To ascertain whether both of these donor populations organoids cluster in relation to each other as well as iP organoids, the clustering analysis should be done in one combined iteration.

As the Southern blot is not available, the authors should provide additional supporting data, such as MyrB and/or CsA treatment (confirming NTCP-dependency) as well as HBeAg measurement (a better de novo infection marker than the virion-associated HBcAg and HBsAg) of infected organoids.

---

## [Author Response]

Essential revisions:All three reviewers recognize the novelty and potential impact of the described liver organoid system on HBV research. Revisions are outlined to strengthen (1) the HBV virology; (2) the conclusions from the RNA-Seq data; (3) information regarding the organoid system.1) The absolute numbers of cccDNA and intracellular HBV DNA, as well as their ratios, varied significantly among the liver organoids from different donors, which could be due to the batch-to-batch variation of liver organoids or the variations of the detection method. In these regards, the authors need to further characterize the viral nucleic acids (cccDNA, RNA, core DNA) in HBV-infected liver organoids more qualitatively and quantitatively.

Indeed as the reviewers point out we observed significant variability in the readouts between the different donors. As the reviewers suggest, to better characterize whether the source of variability of HBV DNA, both intracellular and cccDNA, is due to donor variability, batch to batch variation or variations in the detection method, we quantitatively examined the viral nucleic acids following ex vivo HBV infection of organoids derived from four independent donors. Organoids from the four donors were infected simultaneously with the same batch of virus and intracellular HBV DNA was quantified and detected in parallel to eliminate the contribution of variations in detection method to the observed variability (Lines 135-137 and Figure 1, Figure supplement 3c of the revised manuscript). In order to assess the effect of batch-to-batch variation on the observed variability of infection levels, we also infected independent donors at two different culture passages with the same batch of HBV and we observed comparable levels of intracellular HBV DNA in organoids derived from the same donor at different passages (Lines 135-137 and Figure 1, Figure supplement 3d of the revised manuscript). Our data thus demonstrate that batch to batch variation does not significantly contribute to the observed variability. We did however observe significant differences in intracellular HBV DNA levels between different organoid donors which were infected at the same time with the same batch of virus and processed simultaneously (lines 136 and Figure 1, Figure supplement 3c of the revised manuscript), demonstrating that the variability we observe in the efficiency of infection is indeed donor-to-donor variability. Our data is also consistent with donor to donor variability in infection efficiency observed for primary human hepatocytes (Shlomai et. al, PNAS 2014). We now discuss these results in lines 135-137 of the revised manuscript.

To unambiguously show successful infection of the organoids, detection of cccDNA by Southern blotting is the gold standard and should be shown. Alternatively, a northern blot of HBV RNA or Southern blot of HBV core DNA with and without Nuc treatment can be acceptable.

As the reviewers point out the presence of either cccDNA or HBV core DNA has been demonstrated using Southern blotting on serum from HBV infected patients, on cell-lines containing stably integrated HBV genome (e.g. HepG2.2.15), cell-lines transfected with an HBV genome containing plasmid, or NTCP overexpressing HepG2 cells and iPSC-derived hepatic-like cells infected with HBV, which all are amendable to large scale culture as well as efficient isolation of HBV core particles [e.g. Zhou et al., Antiviral Res. 2006; 72:116-124; Cai et al., Methods Mol Biol. 2013; 1030:151-161; Santantonio et al., Rec Clin Lab 1990; 20(1): 29-35; Bonino et al., Hepatology 1981; 1(5): 386-391; Guo et al., Scientific Rep. 2016; 6:2552; Cui et al., J. Virology 2013; 87(21): 11494-11503; Lee et al., Virus Genes 2003; 27(3): 227-235; Konig et al., J Hepatology 2019; 71(2): 289-300; Sakurai et al., Scientific Rep. 2017; 7: 45698].

Expansion of liver organoids in large scale 3D cultures, their differentiation and ex vivo infection, aside from being time consuming and expensive, presents significant challenges to obtaining large amounts of intracellular HBV core DNA. Indeed a recent paper which described a 3D hiPSC liver organoid system as a model to study HBV infection demonstrated the presence of intracellular HBV core DNA using PCR and not Southern blotting (Nie et al., EBioMedicine 2018; 35: 114–123).

We put substantial effort into setting-up a Southern blot experiment for the ex vivo HBV infected organoid model, which we detail below. However, we came to the conclusion that the scale (100-10000 wells of organoids per condition) and cost of such an experiment (>>€7000,- per condition) precluded its feasibility. Below we will explain in detail how we came to this conclusion.

First we determined the limit of detection for a Southern blot assay and set-up a dot blot assay using biotinylated PCR generated probes spanning either a ~400bp region of the HBV core region or the full-length HBV genome and detected signal using IRDye 800CW Streptavidin using a Li-COR Odyssey Infrared Imager and obtained a specific signal with a linear limit of detection of around 1^10^6^ copies of cloned HBV DNA (Author response image 1). This is in line with limits of detection previously reported for Southern blot (dot blot) detection using either biotin- or 32P- labelled probes [e.g. Saldanha et al., J Virol Methods 1987; 16(4): 339-342; Roe et al., Korean J intern Med 1988; 3(1): 9-14; Santantonio et al., Rec Clin Lab 1990; 20(1): 29-35; Valentine-Thon et al., Mol Cell Probes 1991; 5(4): 299-305]. From this experiment we concluded that we need at minimum a total of 1^10^6^ copies of purified HBV DNA for reliable detection of intracellular HBV core DNA and preferably 1^10^7^ – 1^10^8^ copies to reliably quantify the effect of Nuc treatment.

In order to estimate the amount of organoids needed to perform the Southern blot experiment we performed real-time qPCR on the same amounts of cloned HBV DNA as used in the titration curve we used to assess the detection limit of the dot-blot assay (Author response image 1) and compared these Cq values to Cq values routinely obtained from infected organoid cultures.

When isolating total DNA from HBV infected organoids cultured in one well of a 24 well plate we obtained, depending on the efficiency of infection, Cq values of 23-25 which corresponds to approximately 1^10^3^ – 1^10^4^ copies of intracellular HBV core DNA. This means that in order to obtain a reliable signal in our blot experiment we would need in the range of 100 (bare minimum) to 10000 (optimum) wells of infected organoids for one single condition (with or without Nuc treatment).

**Author response image 1. sa2fig1:** Detection limit of Southern Blot analysis of HBV genome. A. Serial dilution of cloned HBV genome (top row) and non HBV plasmid (bottom row) was spotted on Hybond-XL membrane and detected using a PCR generated biotinylated near full length HBV probe. Signal was detected using IRDye 800CW Streptavidin on a Li-COR Odyssey Infrared Imager. B. The same serial dilution of cloned HBV genome was analyzed by qPCR. Cq values obtained from DNA isolations corresponding to a single well of a 24 wells plate from six independent infections of different donor organoids are indicated by color coded **+**.

Generally for HBV virus production we collect 500ml of supernatant which after concentration is used to infect up to a total of 50 wells (of a 24 well plate) containing liver organoids. Therefore, in order to infect the amount of organoids needed for a Southern blot experiment we would have to collect and concentrate up to 50 liters of supernatant.

Importantly, during the total DNA extraction procedure cellular gDNA (~1ug) is isolated as well as intracellular HBV core DNA. Therefore we applied an existing protocol for HBV nucleocapsid isolation commonly used for cell lines and sera [e.g. Guo et al., J Virol 2005;79(5):2729-42] to our liver organoids and found that this resulted in a ~50% loss of intracellular HBV core DNA as compared to a total DNA extraction protocol while it still contains a high genomic DNA background (100ng genomic DNA/0.005pg (1250 copies) HBV DNA). This is probably due to the intricate three dimensional organoid structure that hampers complete cell lysis in mild lysis buffers. A consequence of this high genomic DNA background is that if we load 1^10^6^ copies of purified HBV DNA we also load ~100ug of genomic DNA which exceeds the binding capacity of the HybondXL membrane (42ug for a 3mm diameter well).

However, to strengthen the HBV virology presented in the manuscript and to further characterize the ex vivo HBV infection of liver organoids, we have performed additional experiments using T5 exonuclease-mediated digestion, a more robust PCR-based technique to detect intracellular cccDNA in four additional donor organoid lines. Using this technique we confirm that the liver organoids are indeed infected with HBV and produce cccDNA (Figure 1f of the revised manuscript). This data, together with extensive immune-fluorescent characterization of expression of HBV proteins HBx and HBV core, as well as characterization of HBV intracellular RNA, DNA, as well as organoid culture supernatants containing infectious HBV particles provides unambiguous data that demonstrates successful infection of the liver organoids.

2) It would be critical to include organoids from cirrhotic donors who have not been previously infected in the analysis. Comparing infected versus uninfected "healthy" individuals makes interpreting the RNAseq data (Figure 4F) ambiguous as it remains unclear what weight the "cirrhotic" gene expression signature carries in the interpretation of results versus HBV infection.

Whether the iP-signature observed corresponds to HBV infection or is a consequence of cirrhosis is indeed a very important point that following the reviewers’ comments, we have made substantial effort to address by including additional donors and patient-derived organoid lines in our RNA seq analysis (figure 6). Specifically, in addition to the healthy infected donor presented in the first version of the manuscript, we now include two additional donors that had previously been infected with but cleared HBV-infection (iDonors). Importantly, these donors had non-cirrhotic livers which were used for transplantation in HBV infected recipients (whose cirrhotic livers were explanted). We also include two donors who had a cirrhotic liver phenotype and underwent explant because of non-alcoholic steatohepatitis (NASH), but were uninfected with HBV (CN). We also included a sixth healthy donor line (hD22) (Lines 309-311 and Figure 6a-d of the revised manuscript). Organoid lines were seeded from liver biopsies or explants from these donors/patients and used in RNA seq analysis. When applying the iP- signature we generated from comparing the five healthy donor organoid lines (hD) to five HBV-infected cirrhotic patient-derived organoid lines (iPs), the iDonors as well as the iP-CN clustered away from the healthy donors (Figure 6a-d). Interestingly, while the iDonors grouped together closely with the iPs (Figure 6a), the CN samples clustered away from the iPs (Figure 6c). These results demonstrate that the iP- signature we obtained is an indicator of HBV infection, and not of the cirrhotic state per se: the HBV infected iDonors organoids, while clustering together with HBV infected iPs, were not generated from cirrhotic livers, and therefore the signature cannot be a cirrhotic signature. These findings are now discussed in lines 295-318 of the revised manuscript.

Also, were input samples analyzed, and are these transcriptome differences consistent with the liver samples before organoid formation? Several genes related to early HCC were up-regulated in HBV patient-derived liver organoids systems. However, the authors did not verify these genes and their expression levels in the HBV-associated HCC patients and HBV infected human liver organoids systems.

Following the reviewers’ suggestion, in order to validate that the transcriptome differences we observed are consistent with liver samples before organoid formation, we examined the expression of candidate genes identified in our gene signature in primary hepatocytes obtained from tissues from healthy or HBV-infected donors. We indeed observed upregulation of signature genes *ADAMTS1, CORO1A, PTHLH, STMN2* and *STY5* mRNA expression in iP primary hepatocytes as compared to hD hepatocytes (Figure 5c). Additionally, we examined expression of the select candidate upregulated signature genes in independent hD organoid lines infected ex vivo with HBV or heat-inactivated virus. In response to HBV infection of organoids, signature genes were modestly upregulated, although not significantly, given the low percentages of infection as well as donor to donor variability inherent to the system (Figure 5 Supplement 1a). We discuss these results in lines 282-289 of the revised manuscript.

3) The authors have not stated how long NTCP-overexpressed human liver organoids and HBV patient-derived liver organoids systems can be kept in culture while maintaining liver cell properties. The authors should confirm these.

Healthy donor, patient-derived and lentivirally transduced NTCP overexpressed liver organoids exhibited liver properties after seven days of culturing in differentiation media (DM). The differentiated hepatocyte markers, which indicate liver cell properties remain stably expressed in DM organoids for the duration of the cultures, which as we have shown in Figure 1—figure supplement 2, can be maintained for more than one month (27 days after infection) (line 132-134). In the absence of ex vivo spin infection, differentiated organoid cultures can be maintained for up to two months with intermittent culturing in expansion media (data not shown). We now provide this information in the methods section (lines 588-590).

Could the authors explain more on why HBV-positive liver organoids were not obtained from HBV patient livers? Have the authors detected HBV integration in the established organoids derived from HBV patients?

As the reviewers point out, we were unable to find any quantifiable signs of HBV replication in the patient-derived organoids. iP- organoid culture supernatants were devoid of HBV DNA, in either expansion or differentiation media. We were also unable to detect intracellular HBV RNA or DNA from iP organoids. Liver organoids are generated from adult stem cells present in the liver. Although these ASCs may not support active viral replication, they may still contain portions of integrated HBV DNA with as of yet uncharacterized function. Previous genome wide analyses in tissues from patients with HCC, had identified regions of the HBV genome to be integrated in both non-tumor HBV infected tissue as well as HCC obtained from these patients (Sung et al., 2012; Zhao et al., 2016). Therefore, in order to better address the reviewers’ comments, we conducted a PCR analysis on genomic DNA obtained from infected patient-derived organoids to examine if HBV is integrated in the genome of these iP lines. Interestingly, in five out of six iP organoid lines analyzed, we found parts of the HBV genome integrated, corresponding to a portion of the HBx coding region (Figure 4—figure supplement 1). Interestingly, in a healthy donor who was previously infected with HBV but since cleared the infection (iDonor1), HBV integration corresponding to the HBx region was also observed (Figure 4—figure supplement 1). This finding indicates that although an individual may have cleared active HBV replication, parts of the genome remain integrated in the host DNA, for which the functional consequences remain to be characterized. This new data is now presented in Figure 4—figure supplement 1 and is discussed in lines 243-250 and lines 406-417 of the revised manuscript.

[Editors' note: further revisions were suggested prior to acceptance, as described below.]

The manuscript has been improved but there are some remaining issues that need to be addressed, as outlined below:The authors add new data to address some of the areas highlighted in the previous peer review including critically important RNA-seq analysis of non-cirrhotic donors and analysis of integration events in iP organoids. The authors also address several points regarding HBV virology, including elimination of the CMV-driven 1.3xHBV construct data.A critical issue that the authors need to address is the hierarchical clustering analysis in Figure 6. The authors provide two analyses (Figure 6A and 6C) which share the iP organoid lines but not the organoids from donors that cleared infection (iD-B) and donors that had NASH (CN). To ascertain whether both of these donor populations organoids cluster in relation to each other as well as iP organoids, the clustering analysis should be done in one combined iteration.

We thank the reviewers for this suggestion. In Figure 6C and 6D of the revised manuscript we now include donors that cleared infection (iDonors 1-4) along with donors that had liver cirrhosis due to NASH (CN1-4) in the same hierarchical clustering analysis and MDS plot. We have made substantial effort to strengthen our analysis and conclusions by including, as stated above, an additional infected donor that cleared HBV (iDonor 4) and two additional donors who had cirrhotic phenotypes due to NASH (CN3-4). By applying the HBV driven iP signature that we generated from comparing the five healthy donor organoid lines (hD) to five HBV-infected cirrhotic patient-derived organoid lines (iPs), we observed that iDonors cluster closely with iPs and away from hDs even though they phenotypically resemble hDs (Figure 6A-B). importantly, inclusion of the iDonors into the analysis, did not change the clustering of the CN samples separately away from the iPs (Figure 6 C-D). These data, generated from additional donors are in agreement with our previous data and support the notion that the iP signature we generated is HBV-driven and not a general cirrhotic signature (lines 313, 320, Table -1).

As the Southern blot is not available, the authors should provide additional supporting data, such as MyrB and/or CsA treatment (confirming NTCP-dependency) as well as HBeAg measurement (a better de novo infection marker than the virion-associated HBcAg and HBsAg) of infected organoids.

Based on the recommendations of the reviewers, to confirm NTCP dependency of HBV infection of the liver organoids, we performed the in vitro HBV infection experiments with or without (pre)treatment of liver organoids with myrcludex-B, the HBV entry inhibitor, which functions by blocking NTCP. Indeed, blocking NTCP results in a reduction of HBV infection as measured via quantitation of intracellular HBV RNA, as well as HBV DNA and HBeAg in the supernatant of infected organoids (presented in Figure 3 C, D and E of the revised manuscript (lines 190-197)). We also measured, as suggested by the reviewers the presence of HBeAg in the culture supernatants of infected liver organoids as an additional marker of active HBV replication (confirming our data demonstrating intracellular HBV RNA and DNA, HBV DNA and Surface Ag produced and detected in the culture supernatant, and HBV viral proteins detected by Immunofluorescence microscopy). Data demonstrating HBV infection mediated production of HBeAg in infected liver organoids are presented in Figure 1G of the revised manuscript (lines 122-123).